# ON PRE-TRAINED LANGUAGE MODELS FOR ANTIBODY

**Danqing Wang**[1,2]\***, Fei Ye**[1]**, Zhou Hao**[3]
[1]ByteDance Research, Shanghai, China      [2]University of California, Santa Barbara
[3]Insititute for AI Industry Research, Tsinghua University
danqingwang@ucsb.edu,  yefei.joyce@bytedance.com
zhouhao@air.tsinghua.edu.cn

## ABSTRACT

Antibodies are vital proteins offering robust protection for the human body from pathogens. The development of general protein and antibody-specific pre-trained language models both facilitate antibody prediction tasks. However, there have been limited studies that comprehensively explore the representation capability of distinct pre-trained language models on different antibody tasks. To investigate the problem, we aim to answer several key questions in this paper, such as how pre-trained language models perform in antibody tasks with different specificity and how introducing specific biological mechanisms to the pre-training process can benefit the model. Additionally, we evaluate if the learned antibody pre-trained representations can be applied to real-world antibody problems, like drug discovery and immune process understanding. Previously, no benchmark available largely hindered the study to answer these questions. To aid in our investigation, we provide an **An**T**ibody **U**nderstanding **E**valuation (`ATUE`) benchmark. We comprehensively evaluate the performance of protein pre-trained language models by empirical study along with conclusions and new insights. Our `ATUE` and code are released at https://github.com/dqwang122/EATLM.

## 1 INTRODUCTION

Antibodies are a type of protein that is useful for diagnosing and treating a variety of diseases, including SARS-CoV-2 (Zhu et al., 2022). It is crucial to understand the information contained in antibody sequences to develop effective therapeutic antibodies and advance our understanding of the immune system (Greiff et al., 2020; Lu et al., 2018; Yermanos et al., 2018). Recent advances in general **P**re-trained **P**rotein **L**anguage **M**odels (PPLM) and specific **P**re-trained **A**ntibody **L**anguage **M**odels (PALM) offer new possibilities for antibody-related tasks. For example, PPLMs have shown promising results in transferring learned representations to antibody tasks (Kim et al., 2021; Zaslavsky et al., 2022) and PALMs have been found to improve model performance in antibody paratope predictions (Leem et al., 2022).

Despite these successes, few studies have thoroughly examined the capability of different pre-trained language models (e.g. general PPLMs and specific PALMs) on various antibody tasks, which hinders the development of better architectures for antibody discovery and modification. To investigate this problem, we compared the performance of the pre-trained protein language model ESM (Rives et al., 2021), the pre-trained antibody language model AntiBERT (Leem et al., 2021), a pre-trained antibody language model EATLM by introducing antibody specific mechanisms, and a model trained from scratch (No Pretrain) on three antibody tasks with varying levels of specificity. The result is illustrated in Figure 1. Here,

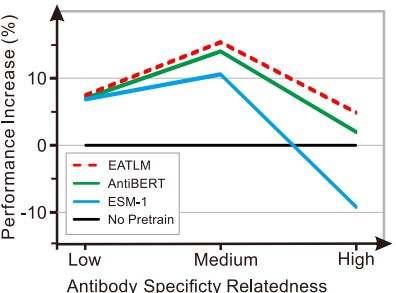

Figure 1: Performance of pre-trained language models on tasks with different antibody specificity.

---

*Work was done when Danqing Wang was in Bytedance Research.

specificity refers to the antibody's unique evolution processes distinct from that of protein to obtain functionality, such as the ability to bind antigen (The definition is discussed in detail in §3.1).

We can see that while ESM performs well in tasks that are less antibody specific, its performance decreases significantly in tasks that are more specific. Additionally, AntiBERT does not demonstrate a clear advantage over the non-pre-trained model in the high-specificity task. These results highlight the limitations of current pre-training language models for antibody-related studies. Using general PPLM representations directly may harm performance, and current pre-training strategies for PALMs may not fit the specific biological functions of antibodies. This emphasizes the need for a comprehensive model design guideline for various antibody tasks. Our main focus is to address the following questions:

*(I) How well will pre-trained language models perform on antibody tasks with varying specificity?* Addressing of the question is mainly hindered by two challenges: the lack of a reliable antibody-specific benchmark for performance evaluation and comprehensive studies of current PPLMs and PALMs. *(II) Can incorporating biological mechanisms, specifically antibody-specific evolution, into the pre-training process provide additional benefits for antibody representation learning?* This idea has been explored in several computational biology studies, which have demonstrated promising results in antibody-related tasks such as disease diagnosis and therapeutic antibody development (Yermanos et al., 2018; Miho et al., 2019). Then, it is interesting to know whether antibody representation learning can benefit from the incorporation of antibody-specific evolution information. *(III) Are the pre-trained antibody representations useful in practical applications, such as drug discovery and immune process understanding?* Antibodies are critical in drug development, and it is essential to determine whether pre-training representations can be beneficial for biologists to comprehend antibody functions or develop drugs.

To investigate these questions, we first propose antibody study benchmark **AnTibody Understanding Evaluation** (**ATUE**). This is the first antibody benchmark with four real-world supervised tasks related to therapeutic antibody engineering, B cell analysis, and antibody discovery. These tasks cover a range of specificity levels to evaluate models on different aspects of antibody biological functions. Based on `ATUE`, we conduct empirical studies to investigate the representation ability of distinct pre-trained language models. To explore the impact of incorporating specific biological mechanisms in antibody pre-training, two objectives are introduced to tailor masked language modeling for evolution: (1) **Ancestor germline prediction** guides the model to discriminate the evolutionary relationship between antibody and ancestral sequences. (2) **Mutation position prediction** mimics hypermutation during the evolution. These methods are used to investigate the representation ability of antibody evolution-tailored language model. Finally, we take a close look at the SARS-CoV-2 antibody discovery to investigate the pre-trained representation under a real-world scenario.

We have three main contributions in this study:

- We created the first comprehensive antibody benchmark called `ATUE` to help with antibody application studies, which includes four real-world supervised tasks ranging from low to high specificity. We also introduce two new objectives for antibody pretraining that incorporate antibody-specific evolutionary information.
- We made key observations for providing guidelines for better antibody representation. Firstly, PPLMs perform well on antibody tasks that have a high relationship with structure, but they perform poorly on tasks with high antibody specificity. Secondly, in most cases, PALMs perform as well as or even better than PPLMs with less pre-training data. Thirdly, PALMs can be improved by incorporating the evolution process, but the evolution information from MSAs does not always benefit antibody tasks.
- We identified 11 potential SARS-CoV-2 binders that have highly identical sequences to existing therapeutic antibodies that bind to the virus, which could accelerate real-world antibody discovery.

## 2    RELATED WORK

Our work focuses on researching the effectiveness of protein and pre-trained antibody language models for antibody-specific tasks. Below we review the representative existing methods. We list the details in Table 1.

**Pretrained Protein Language Models (PPLMs)** There is an increasing interest in exploring large-scale language models using protein sequences (Rao et al., 2019; Madani et al., 2020; Meier et al., 2021; Chen et al., 2022). These models have been shown to achieve state-of-art capacity in predicting protein structure and function. ProtTrans (Elnaggar et al., 2021) and ESM-1b (Rives et al., 2021) take individual protein sequences as input and adopt Transformer language models for pre-training, demonstrating that self-supervision is a promising paradigm for protein secondary structure, contact, homology predictions, and function prediction. To extract evolutionary information from protein sequences, Rao et al. (2021) proposed the MSA-transformer/MSA-1b model utilizing multiple sequence alignment (MSA) instead of a single query sequence as input. This model is superior to ESM-1b for structure prediction, demonstrating evolution information can benefit protein representation learning. Despite the progress in the field, few studies reported their results on antibody tasks.

**Pretrained Antibody Language Models (PALMs)** Encouraged by the success of PLMs in protein representation learning, series work seeks to learn antibody representations based on sequences of antibodies. AntiBERTy (Ruffolo et al., 2021) proposed the first antibody-specific language model, exploring a Transformer trained on 558M natural antibody sequences in the OAS database. Olsen et al. (2022b) train two language models for antibodies: A heavy chain version Ablang-H and a light chain version Ablang-L. The study reported transfer learning results on restoring missing residues of antibody sequences, which is a task similar to pre-training objectives. AntiBERTa (Leem et al., 2021) train the antibody language model on OAS and finetuning AntiBERTa for paratope position prediction, achieving state-of-the-art performance. Recently, Li et al. (2022) proposed an antibody-specific language model and explored its performance in SARS-CoV-2 antigen binding, showing context-dependent representations of antibody sequences benefit binding prediction.

Table 1: Pre-training language models for protein and antibody. Evolution denotes whether evolutionary-related sequences are used during the pretraining. MLM is masked language modeling pretraining objective. HC, antibody heavy chain; LC, antibody light chain.

| Model | Category | Dataset | Evolution | Objective | Antibody Tasks |
|---|---|---|---|---|---|
| ESM-1 (Rives et al., 2021) | PPLM | UniRef50 (27M) | × | MLM | - |
| MSA-1b (Rao et al., 2021) | PPLM | UniRef50 (26M MSAs) | ✓ | MLM | - |
| Ablang-H (Olsen et al., 2022b) | PALM | OAS (14M HC) | × | MLM | Reconstruction |
| Ablang-L (Olsen et al., 2022b) | PALM | OAS (0.19M LC) | × | MLM | Reconstruction |
| AntiBERTa (Leem et al., 2021) | PALM | OAS (72M) | × | MLM | Paratope Prediction |
| EATLM | PALM | OAS (20M) | ✓ | MLM, AGP & MPP | ATUE |

## 3 FRAMEWORK

In this section, we first give a brief introduction to the antibody and its specific evolution. Then we propose the first antibody-specific benchmark (`ATUE`) composed of four tasks with different specificities. Finally, we implement several PPLMs and PALMs baselines and design an evolution-aware PALM to incorporate the biological mechanism into the pre-training process.

### 3.1 BACKGROUND

**Antibody** Antibodies are vital proteins generated by the immune system to remove harmful foreign pathogens in the human body. they can specifically bind to antigens on the pathogen and recognize it. Antibodies are composed of two identical heavy chains and two identical light chains and form a large Y-shaped structure. Two tips on it contain highly variable loops, called Complementarity Determining Regions (CDR), which function for antigen binding.

**Antibody Specific Evolution** Notably, the antibody evolution process is significantly different from that of proteins, providing a good opportunity for us to investigate the impact of general PPLMs on specific subdomains. To perform its protective function, the antibody sequence undergoes evolution selection to search for optimal patterns that can specifically recognize pathogens (Honjo & Habu, 1985). Deciphering the information stored in antibody sequences may benefit our understanding of disease and accelerate therapeutic antibody development (Greiff et al., 2020; Lu et al., 2018; Yermanos et al., 2018). During evolution, the random recombination of V/D/J-gene segments provides the

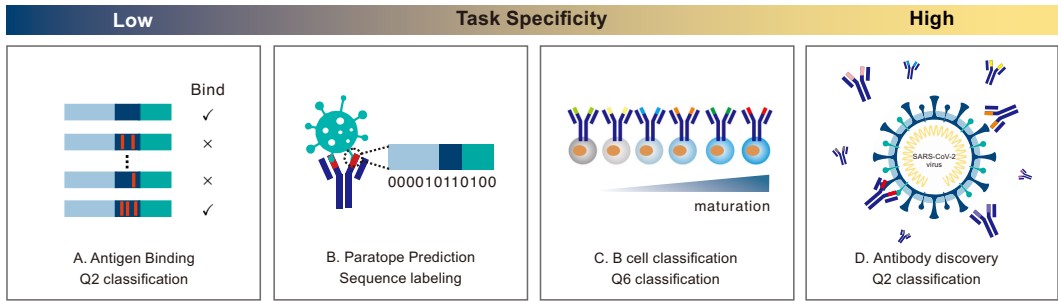

Figure 2: Antibody prediction tasks. The specificity of tasks ranges from low to high.

initial diversity for the ancestor sequence (germline). Upon exposure to a pathogen, this sequence undergoes frequent sequence mutations to search for progeny antibody sequences with optimal binding specificity. In other words, gene recombination provides millions of germlines in the human body, and the germlines further mutate into a huge number of progeny antibodies. Thus, the ancestor relationship between an antibody and its corresponding germline as well as the mutation it undergoes together determine the unique biological functions. In brief, the evolutionary relationships between antibodies arise to gain new functions such as antigen binding. It is significantly different from that of proteins, which are to maintain certain functions across different organisms. We further illustrate this process in Figure 7 in §A.1.

**Unsupervised Antibody Corpus** To obtain the evolutionary information of antibody sequences, we utilize Observed Antibody Space (OAS), a database containing more than 1.5 billion natural antibody sequences (Kovaltsuk et al., 2018; Olsen et al., 2022a) The antibody sequences in the database have been precisely annotated with evolutionary and structural information, including the paired germline and CDR3 for each antibody. To pair the antibody with its germline used in the pretraining task, we used the annotated sequences provided in the OAS database. Further information on data processing can be found in §A.2.

## 3.2 AnTibody Understanding Evaluation (ATUE)

We provide four biologically relevant downstream prediction tasks to serve as antibody benchmarks, covering four major application aspects for antibodies in the real world: therapeutic antibody engineering, disease diagnostics, antibody discovery, and B cell maturation analysis. The antibody specificity of these tasks ranges from low to high, offering scaled tasks with subdomain specificity for pre-trained language model evaluation. Detailed information is listed in Figure 2. All data are publicly open and used under the right license. For each task, we focus on the following aspects and leave the details in Appendix (§A.3 and §A.4):

**[Definition]** The formal definition of the task and the understanding ability required.
**[Impact]** The importance of the task in the biological area.
**[Dataset]** The data source and size.
**[Specificity]** Antibody's specific evolution characteristics are different from general proteins.

We use several classification metrics to evaluate the performance. Accuracy (**ACC**) calculates the ratio of correct predictions. Matthews Correlation Coefficient (**MCC**) is the coefficient between true and predicted values. **F1** is the average weighted score of precision and recall. **AUC** is the area under the ROC curve, which shows the performance at all classification thresholds.

**Antigen Binding Prediction** is a binary sequence classification task to determine whether the CDR region of the antibody can bind to the specific antigen.

**[Impact]** A better understanding of the binding affinity between antibodies and antigens can accelerate the affinity optimization of therapeutic antibodies.
**[Dataset]** We collect the antigen binding data from Mason et al. (2021) and follow the training/validation/test split of 15,128/3,242/3,242.
**[Specificity]** Low. All the antibodies sequence in the dataset are derived from a single germline sequence indicating the task is not antibody-specific evolution-related.

**Paratope Prediction** It is to identify binding positions on the antibody sequence, which is a sequence labeling task to predict a 0/1 label for each residue of CDR fragments.

> **[Impact]** The exploration of paratope (binding positions between antibody and antigen) can help to understand the binding mechanisms of therapeutic antibodies.
> **[Dataset]** The paratope data is collected from Liberis et al. (2018) with 1,662 CDR segments on 277 antibodies.
> **[Specificity]** This task is medium specificity related because only partial antibodies from the database are derived from evolution.

**B Cell Maturation Analysis** It is a 6-category classification task to distinguish the maturation stage of B cell antibody sequences. Each sequence belongs to one of {*immature, transitional, mature, plasmacytes, memory IgD+, memory IgD-*}. It requires the model to learn a representation sensitive to different maturation states.

> **[Impact]** It benefits the understanding of the mechanism during immune evolution, which is a critical biological process in the immune system affecting the function and antigen specificity of antibodies (Ghraichy et al., 2021; Meffre et al., 2000).
> **[Dataset]** We collect 88,094 sequences from Mroczek et al. (2014) with 6 maturation stages.
> **[Specificity]** High. Antibody evolution is highly coupled with B cell maturation (Meffre et al., 2000).

**Antibody Discovery** The task is a binary sequence classification task to distinguish which antibody is directly responsible for SARS-CoV-2 binding. The task is highly challenging from two aspects: (1) Less than 1% of antibodies from SARS-CoV-2 patients are directly responsible for virus binding. (2) It is hard to get a reliable sequence-level classifier using unreliable and noisy individual-level labels.

> **[Impact]** Antibody discovery from B cell repertoire has been widely recognized as a important approach to accelerate antibody discovery for diverse antigens (Weiner, 2015; Pedrioli & Oxenius, 2021), and achieved great success for SARS-CoV-2 antibody discovery (Kovaltsuk et al., 2018; Cao et al., 2020; Shiakolas et al., 2022).
> **[Dataset]** We collected antibody sequences from 133 SARS-CoV-2 patients and 87 health persons from OAS and followed the processing pipeline of Kim et al. (2021). Inspired Zaslavsky et al. (2022), we match the high-ranked sequences with the sequences in the CoV-AbDab (Raybould et al., 2021) database, which have been proved to bind SARS-CoV-2 using wet-lab experiments.
> **[Specificity]** High. It is widely reported antibodies derived from the same disease such as SARS-CoV-2 share strong convergent germline signals (Galson et al., 2020).

### 3.3 EXPERIMENT SETUP

Based on the antibody benchmark `ATUE`, we evaluate the performance of current pertaining language models in different specificity tasks. Furthermore, to investigate the benefit of introducing the biological mechanism, we incorporate evolution information as the extra pretraining objectives for PALMs and propose `EATLM`. The detailed description of the objective and the implementation can be found in §A.5

**Current Pre-trained language models** Existing antibody and protein language models are summarized in Table 1. Since the code and pre-training data of AntiBERTa are not released, we train a BERT model named **AntiBERT** on the full OAS database following the same setting as the original study. **MSA-1b** (Rao et al., 2021) takes protein-specific evolutionary sequences (Multiple Sequence Alignment, MSA) as the input. Because it is hard to align sequences between antibodies due to the diversity of CDR3, we take the germline and create pseudo-MSAs with depth 2. We add a linear layer on top of the language models and finetune the whole model on the downstream tasks.

**Evolution-aware antibody pretraining method** To incorporate the biological mechanism into the pre-training, we propose a model with evolution information: **A**ntibody **E**volu**T**ion-aware pretraining **L**anguage **M**odel. The antibody can be represented as $A$ and the germline of the individual antibody can be represented as $G$. Typically, PALMs are trained with basic masked language modeling (MLM). Based on it, we design another two pre-training objectives to simulate the biological mechanism of antibody evolution. The evolutionary relationship between the antibody and its germline includes two folds: (i) *Whether the antibody and the germline have an evolutionary relationship.* (ii) *How*

Table 2: Performance of PPLMs and PALMs on antibody tasks with increasing specificity. The reported results are the average of repetitive experiments with the standard derivation. EATLM w/o AGP indicates that we remove AGP objective from the pre-taining phase.

| | Antigen Binding (Low) | | | Paratope (Medium) | | | Cell (High) |
|---|---|---|---|---|---|---|---|
| | AUC | F1 | MCC | AUC | F1 | MCC | ACC |
| non-pretrain | 0.858±0.014 | 0.584±0.330 | 0.432±0.183 | 0.845±0.014 | 0.605±0.033 | 0.463±0.037 | 0.554 ± 0.042 |
| ESM-1 | 0.917±0.001 | 0.854±0.002 | 0.689±0.002 | 0.886±0.009 | 0.669±0.024 | 0.547±0.026 | 0.503 ± 0.031 |
| MSA-1b | 0.921±0.001 | 0.857±0.004 | 0.689±0.014 | 0.887±0.009 | 0.679±0.019 | 0.557±0.025 | 0.416 ± 0.050 |
| Ablang-H | 0.918±0.001 | 0.861±0.003 | **0.704±0.010** | 0.878±0.009 | 0.674±0.018 | 0.546±0.023 | 0.570 ± 0.010 |
| Ablang-L | 0.917±0.002 | 0.856±0.001 | 0.682±0.001 | 0.882±0.010 | 0.680±0.018 | 0.553±0.023 | 0.546 ± 0.008 |
| AntiBERT | 0.918±0.003 | 0.843±0.009 | 0.678±0.008 | 0.879±0.011 | 0.690±0.020 | 0.559±0.026 | 0.565 ± 0.028 |
| EATLM | 0.922±0.004 | **0.862±0.004** | 0.699±0.010 | **0.887±0.008** | **0.698±0.017** | **0.575±0.024** | **0.581 ± 0.005** |
| EATLM w/o AGP | 0.920±0.003 | 0.855±0.000 | 0.697±0.008 | 0.883±0.011 | 0.676±0.021 | 0.552±0.027 | 0.559 ± 0.012 |
| EATLM w/o MPP | **0.923±0.001** | 0.855±0.002 | 0.687±0.005 | 0.883±0.011 | 0.691±0.022 | 0.566±0.030 | 0.563 ± 0.010 |
| EATLM w/o AGP & MPP | 0.918±0.001 | 0.845±0.005 | 0.681±0.005 | 0.880±0.009 | 0.674±0.018 | 0.552±0.023 | 0.559 ± 0.009 |

*to mutate residues from the germline to get the specific antibody*. Two evolution-related objectives are introduced to solve the above questions: **Ancestor Germline Prediction (AGP)** and **Mutation Position Prediction (MPP)**. For ancestor germline prediction, we substitute the paired germline $G$ with random germline $G'$ in the batch via a probability $p$. The model is made to distinguish the ancestor germline of the antibody by capturing the shared features. To predict mutation position, for each token in the germline $G$, the objective is to predict a 0/1 label for each token to indicate whether this token has been mutated. For the antibody sequence $S$, we mask the mutation position and predict these tokens.

**Hyper-parameters** We use the base Transformer architecture (Vaswani et al., 2017) with 12 layers, 12 heads, and 768 hidden states. For each task in ATUE, we finetune the model with supervised data. We follow the standard split of Antigen Binding Prediction. For other tasks that do not provide a standard split, we use a 10-fold cross-validation. Since our pre-training model learns the representation of the antibody sequence, we expand the CDR fragment to the full antibody by searching the biological database for therapeutic antibody engineering tasks. We also use the same Transformer architecture to train from scratch for each downstream task. This model is indicated as **non-pretrain** since it is not pre-trained on a protein/antibody database.

**Reproduction** We conduct 10-fold validation on paratope prediction, B cell maturation analysis, and antibody discovery. For antigen binding prediction, we conduct three repetitive experiments with different random seeds. We report the average results and the standard derivation.

## 4 RESULTS AND ANALYSIS

In this section, we present the experimental results and analysis for the representation capability of existing PPLMs, PALMs, and the EATLM method with evolutionary incorporation, using ATUE benchmark. Additionally, we summarize our observations aiming to address the problems highlighted in the introduction.

### 4.1 MAIN RESULTS

**Antigen binding** We evaluate the performance PLMs models for antibody binding and paratope prediction, which are less antibody specific. The results in Table 2 indicate that PPLMs and PALMs perform similarly on these tasks, suggesting that PALMs can learn comparable general protein representations to PPLMs. Among different PALMs, Ablang-H outperforms Ablang-L and AntiBERT. It indicates that separate training for heavy and light chain sequences is beneficial for these tasks. Moreover, the introduction of AGP and MPP provides improvement over AUC and F1 metrics.

**Paratope prediction** The results presented in Table 2 demonstrate that for paratope prediction, both PPLMs and PALMs can significantly boost the prediction accuracy over the model with pre-training. However, PALMs do not exhibit a significant advantage over PPLMs. EATLM outperforms

other models, particularly in terms of F1 and MCC, while other models exhibit high recall and low precision, indicating that they tend to predict more residues as binding sites. With the incorporation of mutation residue prediction, `EATLM` can focus on the specific mutated positions adapted to bind with antigen. Among the two PPLMs, MSA-1b outperforms ESM-1 on F1 and MCC, which benefits from the structure information learning from MSAs.

**B Cell Analysis**  In this task, we investigate the ability of different pre-trained language models to distinguish between various B cell mature states during evolution. The findings, as demonstrated in Table 2, indicate that PPLMs are not effective in discerning minor differences between B cell sequences, resulting in mediocre results. Both ESM-1 and MSA-1b perform significantly worse than randomly initialized models. MSA-1b, in particular, performs poorly among all pre-trained language models, implying that representations that excel in protein structure prediction may be detrimental to antibody-specific tasks. Conversely, all PALMs show promising results for the task. This may be due to the fact that the general protein has little correlation with the specific antibody mature process and cannot capture this feature during

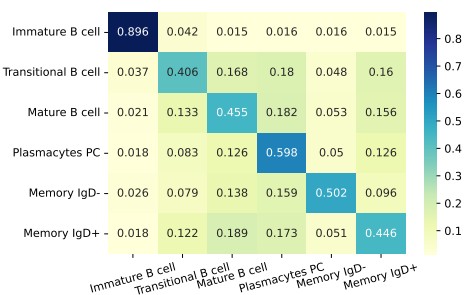

Figure 3: B cell evolution category prediction. For each $p_{ij}$ in $i-$th row and $j$-th column, it means the frequency for the model to predict the antibody in $i$ category to $j$ category. The number is normalized by row.

protein pretraining. Our `EATLM` significantly outperforms the other PALMs. This is because our model can effectively capture the evolution feature and better distinguish between B cells at different stages of maturation by explicitly modeling the biological mechanism.

We conduct further analysis to figure out whether our `EATLM` successfully captures sequence characteristics during the evolutionary process. We explore the probabilities of predicting antibodies in class $i$ to class $j$. The results shown in Figure 3 reveal `EATLM` can easily classify the immature B cell with an accuracy of 0.9. It is consistent with the biological study that CDR3 sequence length in immature B cells is significantly shorter than that of the other mature B cells (Ghraichy et al., 2021). From the diagonal, we can figure out that our model tends to mistake the B cell sequences with their previous or post-evolutionary stage, consistent with the biological process.

**Antibody Discovery**  We investigated the potential of PPLMs and PALMs in aiding the discovery of antigen-specific antibodies for real-world problems. To achieve this, we followed a two-step process similar to Zaslavsky et al. (2022). Firstly, we created a sequence classifier to differentiate SARS-CoV-2 antibodies using noisy individual-level labels. Secondly, we compared the highly-ranked sequences with true binding sequences in the CoV-AbDab (Raybould et al., 2021) database to determine if there are similarities. We used a 90% sequence identity threshold to determine the likelihood of biological functionality similar to the existing binders. The experimental design for this is outlined in §A.7.

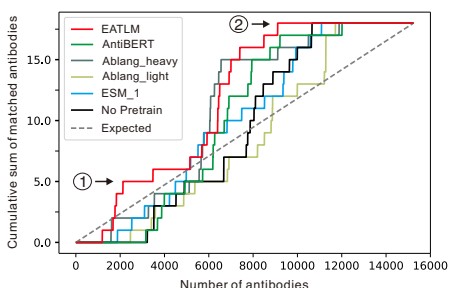

Figure 4: The cumulative sum of matched sequences number in the order of the predicted probability. EATLM outperforms other PALMs and PPLMs for finding SARS-CoV-2 binder faster highlighted in ① and finding all antibodies faster highlighted in ②.

Figure 4 shows the cumulative sum of matched sequences in the order of predicted probabilities by different pre-trained language models for the SARS-CoV-2 specific antibody discovery task. We can observe that PALMs outperform PPLMs in identifying potential binders, as the sequences predicted with high probability by PALMs match better with the existing binders. Moreover, among PALMs, `EATLM` significantly outperforms other models, with the red line indicating its performance. Initially, `EATLM` is the quickest method to find potential binders, but it loses to Ablang-H, and eventually overtakes again and converges. This suggests that `EATLM` is the most effective method for identifying all potential binders in this dataset.

Furthermore, we list 11 potential binder sequences discovered by `EATLM` in Table 3. Without supervised labels, `EATLM` gives a high probability of 2 SARS-CoV-2 existing binding antibodies. Besides, `EATLM` suggests 9 potential sequences with high CDR-H3 sequence identity, indicating the potential for diverse-epitope antibody discovery and selection. These results demonstrate the potential of `EATLM` in therapeutic antibody discovery.

To validate whether the antibody sequences with 90% sequence identity can indeed bind the same target, we investigate the 3D structure of the true binding antibody. Table 4 shows only one single residue difference between the predicted binder and the existing binder, suggesting the predicted binders are highly possible to interact with SARS-CoV-2.

Table 3: The CDR3-H3 region of high-ranked sequences to bind to SARS. We show the CDR3 fragment of the heavy chain in the antibody sequences. 'Identity' is the similarity between the predicted binder and the true binder. The epitope of the true binders is shown. The origin of the majority of the true binder sequences is B cells from patients. The different amino acids between the predicted binder and the existing binder are highlighted in red

| No | Predicted Binder | Existing Binder | Epitope | Identity |
|---|---|---|---|---|
| 1 | AREGIVGATTGFDY | AREGIVGATTGFDY | spike | 1.000 |
| 2 | ARDLGGYFDY | ARDLGGYFDY | RBD | 1.000 |
| 3 | AKDQDDAYYYYYYMDV | AKDQDDGYYYYYYMDV | NTD | 0.938 |
| 4 | ASYYYDSSGYHYGMDV | ASYYYDSSGYYYGMDV | RBD | 0.938 |
| 5 | ARRGLGLYYYGMDV | ARRGDGLYYYGMDV | S2 | 0.929 |
| 6 | ARAFRGSYYYGMDV | ARATRGSYYYGMDV | S2 | 0.929 |
| 7 | ARLSGSSWYFDY | ARLSGSSWDFDY | spike | 0.917 |
| 8 | ARLGSSSWYFDY | ARVGSSSWYFDY | spike | 0.917 |
| 9 | ARGWLRGYFDL | ARRGWLRGYFDL | RBD | 0.909 |
| 10 | ARDWGELYFDY | ARDWGEYYFDY | RBD | 0.909 |
| 11 | ARDLGGVFDY | ARDLGGYFDY | RBD | 0.900 |

Table 4: 3D structure of the true SARS-CoV-2 binding antibody No.3. G2A highlights the single atom difference in No.3, indicating the predicted binder is highly likely to bind the virus.

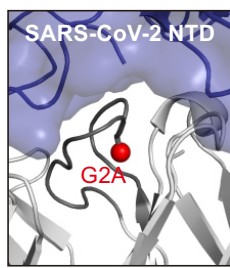

PDB:7N62

## 4.2 HOW DOES EVOLUTION PRETRAINING TASK INFLUENCE THE REPRESENTATION?

To comprehend the reasons for the better performance of `EATLM` on antibody-related tasks, we conduct the analysis of the pre-trained representations. The objective of this analysis is to evaluate the effectiveness of the evolution-aware pre-training strategies from two perspectives: (1) Does the pre-trained representation of antibodies reflect their ancestor relationship? (2) Does the specificity of antibodies get captured by the evolution objective?

**Ancestor Gerlime Visualization**  We perform UMAP visualization analyses in Figure 5. First, we observe that antibodies evolved from the same germline are nicely clustered together (Figure 5a and 5b), indicating the learned embedding is encoded with germline information. Besides, sequences with similar scales of evolutionary distance tend to cluster together, and a clear gradation of evolutionary distance can be observed in Figure 5c and 5d. The visualization provides a sanity check for the ability of EATLM to extract the sequence information of antibodies.

**Accuracy of Mutation Position**  Based on the specific evolution process described in §3.1, we can find the mutation during the evolution process bring specificity to the antibody. Thus, we explore the model's ability to predict mutated residue from the masked token, which can reflect the specificity feature the model captures. We find that although AntiBERT can predict with an accuracy of 0.889 on all positions, it fails on mutation positions with a 0.031 accuracy. In contrast, `EATLM` achieves an accuracy of 0.443 on mutation position, which indicates that the model captures the specificity information. Note that during the MPP training, we mask the mutation position on antibody sequences, which are different from its germline. Thus, the model cannot get the mutated residue from the germline directly. The only way is to learn the underlying mutation rules. The full results are shown in Table 8 in Appendix.

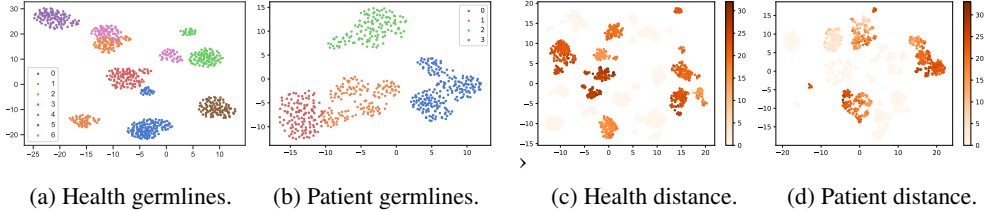

(a) Health germlines.    (b) Patient germlines.    (c) Health distance.    (d) Patient distance.

Figure 5: UMAP Visualization. (a-b) for sequences with different germlines. One color indicates one germline with its descendant. (c-d) for evolutionary phylogeny distance. The shade of color indicates its distance from the ancestor germline.

### 4.3 KEY OBSERVATIONS

**The performance of pre-trained language models is highly dependent on the specificity of the task.** In tasks with low antibody-specificity, PPLMs perform similarly to PALMs, indicating that using general protein representations from PPLMs is an effective way to transfer learning in these tasks. On medium specificity tasks such as paratope prediction, PALMs show their advantage and outperform PPLMs. However, for tasks with high specificity, PPLMs have significantly lower performance, suggesting that general pre-trained protein models are insufficient for antibody-specific representation learning. Additionally, incorporating protein evolution information does not always benefit antibody tasks, especially those that require antibody evolution information, as shown by the 20% decrease in performance observed with MSA-1b compared to the model without pre-training. This finding is consistent with the biological understanding that the mechanism of antibody evolution is significantly different from that of proteins.

**Incorporation of biological evolution mechanism into PALM generally benefits antibody prediction tasks.** The inclusion of evolution-related training objectives assists in identifying mutation positions on antibodies, which is a distinguishing feature from germline. Notably, the performance increase of EATLM in comparison to other PALMs is linked with the level of task specificity. The ablation study showed removing the evolution-related pre-training objectives leads to decreased performance, confirming their contribution to the prediction task. Further research in this direction is promising and could offer more in-depth insights.

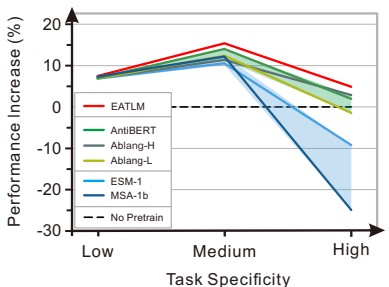

Figure 6: Performance summary of various pre-trained language models.

**Antibody pre-trained representations are helpful for real-world drug discovery.** By utilizing the language model, we predict the likelihood of each antibody binding with SARS-CoV-2. Despite lacking precise sequence-level labels, we successfully identify 11 promising antibody binders.

## 5 CONCLUSIONS AND LIMITATIONS

In this paper, we conduct a detailed investigation into the effects of pre-trained protein and antibody language models on various antibody tasks. To facilitate research in the antibody and machine learning fields, we provide ATUE consisting of four important antibody tasks from four different biological categories with varying levels of antibody specificity.

However, there are certain constraints to our research. Firstly, due to the scarcity of data, the diversity of tasks in our ATUE is limited. As more data becomes available, we anticipate expanding our benchmark to include a greater range of diseases and larger data sets. Additionally, we did not examine any 3D structure information during antibody pre-training. As antibody structures offer more information than just sequences, such as geometry, incorporating structural information in future studies may lead to improved results.

ETHIC STATEMENT

This research involving the use of pre-existing data and computational methods did not involve any human or animal subjects, and therefore, no ethical approval was required. The authors followed all applicable ethical standards and guidelines for data analysis and reporting. All data used in this study were obtained from publicly available sources, and proper citation and attribution have been given. The authors have made efforts to ensure that the research presented in this paper does not infringe upon any existing copyrights or intellectual property rights.

ACKNOWLEDGEMENT

We thank members of ByteDance Research for discussion, Zaixiang Zheng and Yi Zhou for useful writing suggestions. Hao Zhou is supported by Vanke Special Fund for Public Health and Health Discipline Development, Tsinghua University (NO.20221080053), Guoqiang Research Institute General Project, Tsinghua University (No. 2021GQG1012);

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

# A   APPENDIX

## A.1   ANTIBODY SPECIFIC EVOLUTION

Antibodies, composed of two identical heavy chains and two identical light chains, form a large Y-shaped structure, where the two tips are responsible for pathogens binding. Antibody evolution, described by sequence-sequence relationships between ancestor and progeny antibodies, reflects antibodies' key antigen-binding function (Honjo & Habu, 1985). During antibody evolution (Figure 7), the initial diversity is encoded into the ancestor sequence through randomly recombination of V-, D- and J-gene segments. Upon exposure to a pathogen, the sequence undergoes frequent sequence mutations to search for progeny sequences with optimal binding specificity. Sequence evolution analysis has been employed by many computational biology studies and shows promising results in antibody-related tasks, such as disease diagnosis and therapeutic antibody development (Yermanos et al., 2018; Miho et al., 2019).

Importantly, antibody evolution is significantly different from that of proteins. Antibodies only contain hundreds of thousands ancestor sequences so-called germline. To bind dozens of millions of diverse antigens, antibodies need to mutate from the ancestor sequences to gain new functions (Figure 7). Therefore, the **non-conserved amino acids** (mutated ones) plays important roles for structure and function. On the contrary, the **conserved amino acids** (not mutated) in proteins determine structure and function. During protein evolution, evolutionary pressure to maintain protein structure and functions leads to the conservation or co-evolution of residues located in structural folding core for binding interface. Diverse methods have been developed to extract the co-evolution information from conserved amino acids sequences for structure and function prediction, such as AlphaFold (Jumper et al., 2021).

In brief (Figure 7), antibody evolution specificity distinct from that of proteins can be defined with two main features: (i) ancestor germlines; (ii) the mutated amino acids of germlines.

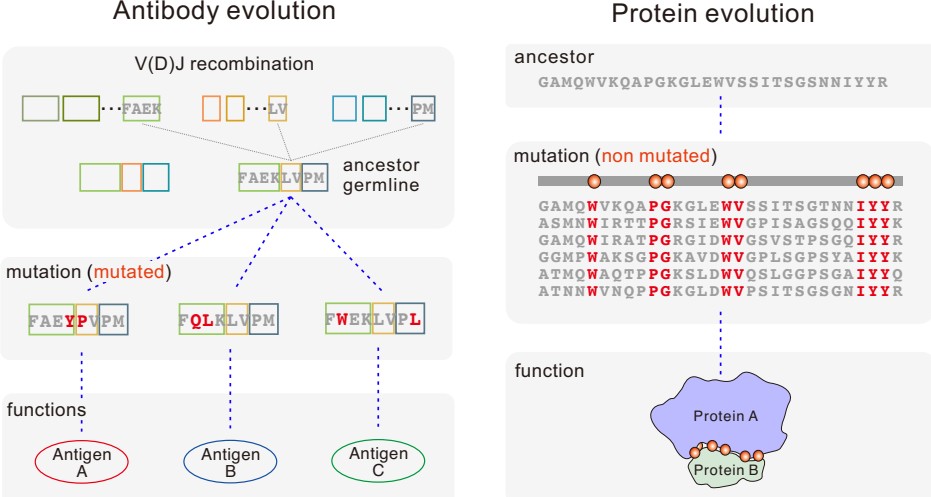

Figure 7: Evolution specificity of B cell antibodies comparing with general proteins. The evolutionary sequence relationships are highlighted using blue dash lines.

## A.2   DATA PROCESSING DETAILS

**Pairing Antibody with Germline**   For germline annotation in the pre-training task, we used the annotated germline sequences provided in the OAS database (Kovaltsuk et al., 2018). For downstream benchmarks tasks like B-cell classification, therapeutic antibody engineering, and disease diagnosis, we completely followed the methods shown in the OAS database paper. IgBLAST, an immunoinformatic benchmarking tool for the analysis of B-cell antibody repertoires was used for germline annotation (Ye et al., 2013). The antibody nucleotide-containing FASTA file was aligned

to germline and translated to amino acids using IgBLASTn. The antibody amino-acid sequence was aligned using IgBLASTp. The germline databases for human patients used ImMunoGeneTics (IMGT) germline sequences derived from Lefranc et al. (1999). For each antibody, usually, multiple germline sequences can be obtained and only the single sequence showing the highest confidence score for the alignment was chosen.

**Pre-training Data Processing**   We downloaded OAS Oct 2021 version from its website and removed duplicate sequences. To avoid data leakage, we cluster sequences based on the CDR3 sequence and filter each cluster by 70% identity over the whole sequence using Linclust (Steinegger & Söding, 2018). Then, we shuffle the dataset and split it into 100k-size chunks. The last chunk is used as the validation set. The dataset size is 20,245,249 and 45,249 are used for validation.

Table 5: ATUE benchmark. The five downstream tasks are divided into three biological categories and each category focuses on different input levels. 'Q6' indicates the classification task have 6 classes. For disease diagnosis, the size indicate the number of profiles, where each profile contains thousands to millions of sequences.

| Specificity | Task Name | Input | Formalization | Size |
|---|---|---|---|---|
| Low | Antigen Binding Prediction | CDR fragment | Q2 Cls. | 21,612 seqs |
| Medium | Paratope Prediction | CDR residue | Q2 Labeling | 1,662 seqs, 21,342 positions |
| High | SARS Antibody Classification | Sequence | Q2 Cls. | 22,000 seqs |
| High | B Cell Classification | Sequence | Q6 Cls. | 88,094 seqs |

## A.3   ATUE DETAILS

We summarize the tasks used in ATUE in Table 5 and discuss each task in detail in this section.

**Antigen Binding**   Accurate antigen-binding prediction approaches could allow significantly more efficient antibody discovery with higher affinity. Machine learning methods have already achieved some success in antibody binding capacity optimization. We collect the antigen-binding data from Mason et al. (2021) and follow the training/validation/test split of 15,128/3,242/3,242. The original dataset only has CDR3 fragments, and we extend them to the full antibody sequences. For cross-validation, we split the dataset by antibody sequences to ensure that no antibody sequences overlap between 90% training and 10% validation.

**Paratope Prediction**   Paratope is the antibody residues involved in antigen binding. The ability to accurately map the paratope can provide detailed knowledge about the binding mechanism and accelerate antibody discovery. 1D sequence-based deep learning methods have been employed for paratope prediction. The paratope data is collected from Liberis et al. (2018) with 1,662 CDR segments on 277 antibodies. Each antibody contains three CDR fragments (CDR1, CDR2 and CDR3) in the heavy chain and three CDR fragments in the light chain. We also search the full sequence for each antibody and use the whole sequence as input. For cross-validation, we split the dataset by antibody sequences to ensure that no antibody sequences overlap between 90% training and 10% validation.

Table 6: The statistics of B cell classification.

| Type | Size |
|---|---|
| Immature b cell | 14,145 |
| Transitional b cell | 13,197 |
| Mature b cell | 16,139 |
| Plasmacytes PC | 22,236 |
| Memory IgD- | 8,437 |
| Memory IgD+ | 13,940 |

**B Cell Analysis**   We formulate a 6-category classification task for B cell maturation analysis, which includes {*immature, transitional, mature, memory IgD+, memory IgD-, plasmacytes,*}. The analysis of B cell maturation plays an important role in understanding the mechanisms underlying B cell responses in the immune system  Ghraichy et al. (2021); Meffre et al. (2000).

The order of B cell type follows the evolutionary process in the immune system, from an immature state to a transitional state, and finally becomes a memory B cell. Both memory IgD- and IgD+ belong to memory B cells with different isotypes, and they have a high affinity to foreign antigens. Among the other categories, the Plasmacytes PC sequences also have some affinity ability. It is widely reported that changes in antibody sequence patterns correlate with B-cell maturation. Therefore, we use this task to evaluate the representation learning capacity of the language model.

We collect 88,094 sequences from  Mroczek et al. (2014). They extracted from the peripheral blood of healthy adults and got six types of B cells with different maturity and antibody sequences. The distribution of various types of B cells in the dataset is shown in Table 6

**Antibody Discovery**   Antibody discovery from B cell repertoire has been widely recognized as a novel trend to improve the efficiency of antibody discovery for diverse antigens  (Weiner, 2015; Pedrioli & Oxenius, 2021). However, previous studies highly rely on expensive wet-lab experiments (Cao et al., 2020; Shiakolas et al., 2022). Deep learning-based methods have shown the potential capacity to help antibody discovery by reducing cost and increasing efficiency (Widrich et al., 2020; Wang et al., 2022). Here, we ask whether pre-trained models can benefit real-world problems and enable fast-track neutralization of SARS-CoV-2 antibody discovery.

In the first step, we develop a sequence classifier to distinguish which antibody sequence from the numerous sequences is responsible for the recognition of SARS-CoV-2. This task is highly challenging since we can hardly get the sequence-level disease label that indicates whether the antibody sequence is related to the disease. Thus, we follow the practice of  Roskin et al. (2020); Zaslavsky et al. (2022) to use the individual label as the rough sequence label and train a sequence-level predictor. Then, with the help of a sequence-level predictor, we can give each sequence a most likely label to help antibody discovery, whose accuracy has been verified by the excellent results on individual prediction, which may accelerate the discovery of new antibody sequences.

We follow the condition of  Kim et al. (2021) to filter SARS-CoV-2 antibody data from the OAS database. The basic condition is '*Chain = heavy; Isotype = IGHG; BSource = PBMC; Species = human; Vaccine = None*'. We further add the condition of '*Unique Sequences >= 10000*'. For health/SARS we set the '*Disease*' field to '*None*', '*SARS-CoV-2*'. Then we obtain 87/133 patient profiles for each type. To make a balanced dataset, we limit the size of the health profile and mix up the healthy ones and the ones with the SARS-CoV-2. For cross-validation, we randomly split the dataset by profiles 10 times: 90% for training and 10% for validation. We further select sequences with top100 redundancy to make the positive labels more accurate.

## A.4  QUANTITATIVE ANALYSIS OF ATUE TASK SPECIFICITY

It is important to include statistical significance tests relative to the antibody-specific features in antibody functional tasks we proposed in the ATUE benchmark. According to the evolution process shown in Figure 7, antibody evolution specificity distinct from that of proteins can be defined with two main features: (i) ancestor germlines; (ii) the mutated amino acids of germlines. We implemented statistical significance tests of (i) ancestor germlines subtype usage; (ii) the number of mutated amino acids in antibodies against different labels of downstream tasks in ATUE to quantitatively assess the "Task specificity". The analysis is now summarized in Table 7. Generally, it is clearly shown that `ATUE` benchmark comprises antibody tasks showing different scales of antibody specificity for later modeling analysis. Moreover, they are used for statistical analysis of task specificity and pre-training model objectives in our study.

**Antigen Binding**   In the Antigen binding dataset, both antibody binding and none antigen-binding sequences share the same germline subtype sequence (IGHV3.1) (Figure 8A) as well as the same number of germline mutations 8B). Therefore, None of two antibody-specific features show significant distribution differences between data with different labels, demonstrating antigen binding is a task with low antibody specificity.

Table 7: Task antibody specificity. Summary of the statistical significance test of two antibody-specific features for different tasks in the `ATUE` benchmark.

| Task | Statistical significance (p-value) | | General Specificity |
|---|---|---|---|
| | Germline Usage | Mutations Numbers | |
| Antigen Binding | Nan | Nan | Low |
| Paratope Prediction | 0.296 | 0 | Medium |
| B cell classification | 0 | 0 | High |
| SARS antibody discovery | 0 | 0 | High |

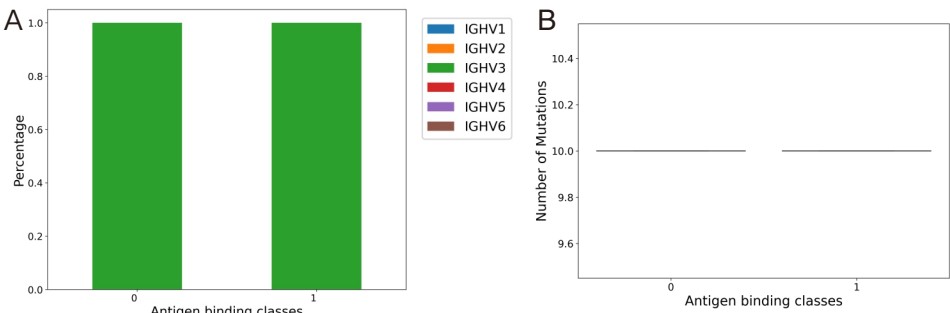

Figure 8: (A)IGHV gene segment usage distribution between binding and non-binding antibody. (B) Germline mutation number distribution between binding antibodies and non-binding antibodies.

**Paratope Prediction**  For the paratope prediction task, we first evaluate the germline subtype distribution difference between sequences with different numbers of binding sites (Figure 9A). Kruskal Wallis test showed a p-value of 0.296 suggesting germline subtype usage is not statistically significant. Also, we find the binding sites can be significantly mapped with more germline mutations than the non-binding sites, which is consistent with the knowledge of antibody specificity definition (Figure 9B). One out of two antibody-specific features shows significant distribution differences between data with different labels. Therefore, we define this task as a medium specificity task.

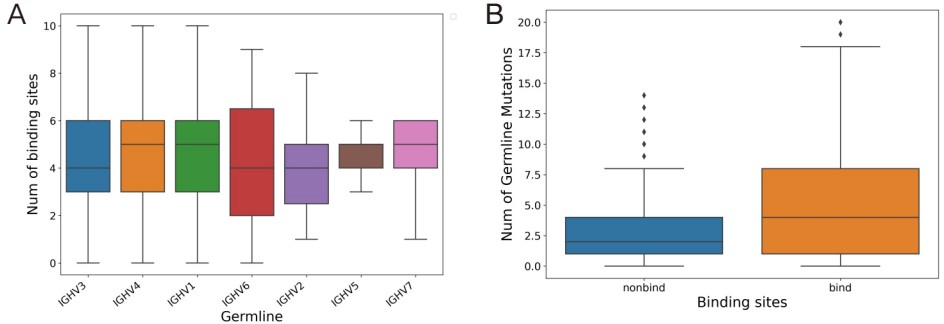

Figure 9: (A) Number of binding sites distribution between different IGHV gene segments. Comparison is performed using the Kruskal-Wallis test with a p-value of 0.296. (B) Germline mutation number distribution between binding and non-binding positions. Comparisons performed using t-tests show p-value equals 0.

**B Cell Analysis**  As shown in Figure 10, the distribution of the germline usage as well as the number of germline mutations are significantly different between antibodies in B cells with different developmental stages. This observation is highly consistent with previous studies Mroczek et al. (2014); Ghraichy et al. (2021). Since both of the antibody-specific features show significant distribution differences, this task is defined as a high-specificity task.

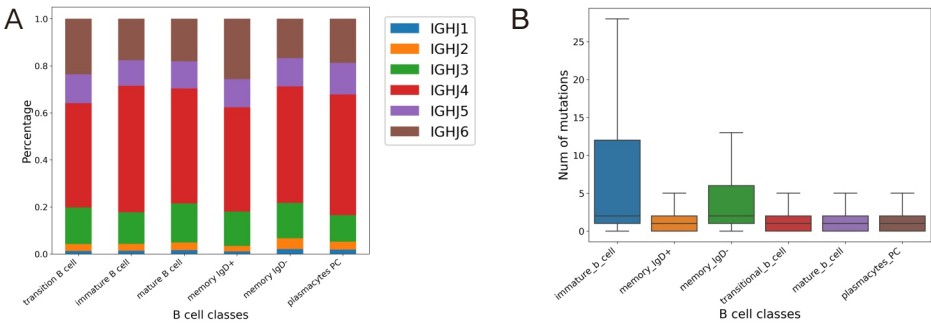

Figure 10: (A)IGHV gene segment usage distribution between different B cells. Comparison is performed using the chi-squared test with a p-value of 0. (B) Germline mutation number distribution between different types of B cells. Comparisons performed using the Kruskal-Wallis test show the p-value equals 0.

**SARS Antibody Discovery**  Antibodies in SARS patients and healthy ones show a significant difference in their germline subtype usage and the number of germline mutations (Figure 11). This observation is highly consistent with previous studies showing SARS antibody convergent among patients Galson et al. (2020). Since both of the antibody-specific features are highly significant, this task is defined as a high-specificity task.

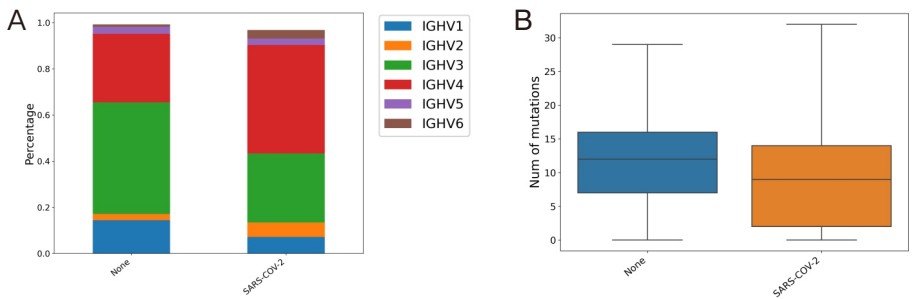

Figure 11: (A)IGHV gene segment usage distribution between antibodies in SARS patients and healthy ones. Comparison is performed using the chi-squared test with a p-value of 0. (B) Germline mutation number distribution between antibodies in SARS patients and healthy ones. Comparisons performed using the Kruskal-Wallis test show a p-value equals 0.

## A.5   MODEL TRAINING DETAILS

Antibody can be represented as $A = \{a_1, a_2, \cdots, a_m\}$ and the germline of individual antibody can be represented as $G = \{g_1, g_2, \cdots, g_n\}$, where $m$ and $n$ are the lengths. Each token $a_i$ or $g_j$ in the sequence is called a residue that belongs to the amino acid set $\mathbb{A}$. $\mathbb{A}$ includes 20 common amino acids

with a residue 'X' that indicates the residue is unknown (mostly in the germline). Typically, antibody PLMs are trained with basic mask language modeling objective $l_{\text{MLM}}$ on the antibody sequences $S = A = \{a_1, \cdots, a_m, \}$.

### A.5.1 EVOLUTION-AWARE PRETRAINING

In order to incorporate the evolutionary information into the pre-training, we pair the antibody sequence $A$ with its germline $G$ and concatenate them into a long sequence with a special token '[SEP]' as the delimiter: $S = \{s_1, \cdots, s_{m+n+1}\} = \{a_1, \cdots, a_m, [\text{SEP}], g_1, \cdots, g_n\}$. Thus, we optimize the MLM objective on the long sequence $S$:

$$l_{\text{MLM}} = -\frac{1}{|M|} \sum_{i \in M} \log p(s_i | S_{\backslash M}), \tag{1}$$

where $M$ is the index set of masked tokens. It helps the model learn the basic residue distribution for antibody sequences. Besides, it can also capture the interaction between residues of the antibody and its germline.

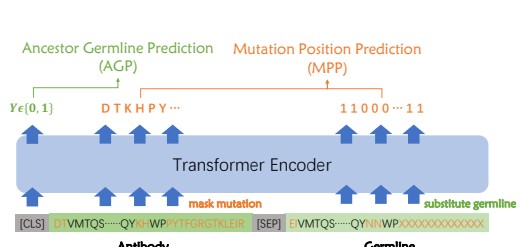 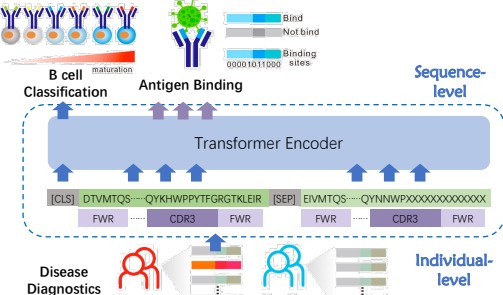

(a) Pre-training with two biological evolution tasks.   (b) Finetuning for three biological categories in ATUE.

Figure 12: EATLM. In Figure 12a, AGP randomly unpairs the germline sentence and predicts the ancestor relationship. MPP predicts the mutation position on the germline and the masked mutation residue on the antibody. Based on the input, the three categories in ATUE can be divided into sequence-level and individual-level (Figure 12b). For individual-level disease diagnostics, we score each sequence in the individual profile and calculate the trimmed mean over all sequences to get the individual score.

**Ancestor Germline Prediction** The ancestor relationship between the antibody and its germline determines the shared biological functions obtained in the evolution. Antibody sequences with similar residues evolved from different germline sequences may have different biological functions. When stimulated by a foreign antigen, the common ancestor germline evolves to various antibody sequences. Similar antibody sequences may have different germline sequences, which will affect their biological functions. Thus, the aim of this task is to determine whether the antibody has an evolutionary relationship with the given germline. During training, we substitute the paired germline $G$ with random germline $G' = \{g_1, \cdots, g_n\}$ in the batch via a probability $p = 0.3$. The new sequence is denoted as $S' = \{a_1, \cdots, a_m, [\text{SEP}], g_1', \cdots, g_n'\}$ and the training loss can be described as:

$$l_a = -\log p(y | S'), \tag{2}$$

where $y \in \{0, 1\}$ indicate whether the noisy germline $G'$ is the ancestor of the antibody $S$. It can help the model to distinguish the ancestor germline of the antibody by capturing the shared features.

**Mutation Position Prediction** The somatic hypermutations on the germline further give progeny antibodies the specificity of binding with the specific antigen. In order to model this specificity, this task focuses on predicting the mutation positions and the residues mutated. Specifically, for each token $g_j$ in the germline $G$, the target is to predict a label $y_j \in \{0, 1\}$ to indicate whether this token has been mutated. For the antibody sequence $S$, we mask the mutation position and predict these tokens. The objective can be formalized as:

$$l_m = -\frac{1}{n} \sum_{j \in \{1, \cdots, n\}} \log p(y_j | S_{\backslash M'}) - \frac{1}{|M|} \sum_{i \in M'} \log p(a_i | S_{\backslash M'}). \tag{3}$$

Here, $M'$ is the ground-truth mutation position and we mask these tokens on the antibody sequence. This task is more difficult than MLM which equally masks tokens in the $L$, because the tokens on the mutation position of $A$ get less information from the germline, compared with other shared residues between the antibody and the germline. By optimizing this objective, the model learns to capture the specificity obtained from the somatic hypermutation in the evolutionary process.

### A.5.2  IMPLEMENTATION DETAILS

We use the base Transformer architecture (Vaswani et al., 2017) with 12 layers, 12 heads, and 768 hidden states. The total parameters are 86M. We use Adam optimizer (Kingma & Ba, 2015) with a maximum learning rate of 2e-4 and a warm-up step of 24,000. The maximum length is set to 400 since most antibody sequences are shorter than 180. We first pre-train our model with the MLM objective. During the pre-training, 15% tokens are randomly selected with 80% masked, 10% replaced, and 10% kept. Then we conduct further pre-training on two antibody-related tasks with a smaller learning rate of 1e-5.

For each task in `ATUE`, we finetune the model with supervised data. We follow the standard split of Antigen Binding Prediction. For other tasks that do not provide a standard split, we conduct 10-cross validation and report the average results. Since our pre-training model learns the representation of the antibody sequence, we expand the CDR fragment to the full antibody by searching the biological database for therapeutic antibody engineering tasks. For finetuning, we limit the max epochs to 30 and use the Adam optimizer with a max learning rate of 3e-5. We use the mean representation of 12 layers as the sequence representation.

Table 8: The pretraining details for different models. The 'Germline' is the prediction accuracy of AGP, and the 'Position' and 'Mutation' are the accuracy of mutation positions and the mutated residues respectively.

| Model | Size | Loss | Step | Germline | Position | Mutation |
|---|---|---|---|---|---|---|
| AntiBERT | 85M | 0.437 | 568000 | / | / | 0.031 |
| AntiBERT w AGP | 85M | 0.558 | 587000 | 0.330 | / | / |
| AntiBERT w MPP | 85M | 1.744 | 622000 | / | 0.965 | 0.410 |
| ESM-1 FT | 85M | 0.2636 | 136000 | / | / | / |
| ESM-1b FT | 650M | 0.2704 | 278000 | / | / | / |
| EATLM w/o AGP & MPP | 85M | 0.143 | 108000 | / | / | / |
| EATLM w/o AGP | 85M | 1.744 | 134000 | / | 1.000 | 0.442 |
| EATLM w/o MPP | 85M | 0.001 | 126000 | 1.000 | / | / |
| EATLM | 85M | 1.856 | 252500 | 0.996 | 1.000 | 0.443 |
| EATLM-large w/o AGP & MPP | 650M | 0.149 | 193000 | / | / | / |
| EATLM-large w/o AGP | 650M | 1.753 | 267000 | / | 1.000 | 0.438 |
| EATLM-large w/o MPP | 650M | 0.001 | 200000 | 1.000 | / | / |
| EATLM-large | 650M | 1.773 | 384000 | 0.996 | 1.000 | 0.431 |

The model is trained with 108,000 steps and gets a 0.9606 token accuracy on the MLM task. It takes further steps for AGP and MPP. The model quickly converges for AGP and gets a 0.99 accuracy on the ancestor germline prediction because more than 80% residues are shared between the antibody and its germline. For MPP, it can predict the mutation position with the accuracy of 1.000 and obtains a 0.442 accuracy in the mutation position (`EATLM` w/o AGP). It means that the model can easily find the mutation positions by the self-attention between the antibody and germline, but it is still difficult to predict which residues this position will mutate to. We assume it is because the ancestor germline can undergo different somatic hypermutations and get various progeny antibodies, resulting in different valid mutations at the same position. We also compare this mutation accuracy with the model without MPP, which is only trained with MLM on the concatenation of the antibody and its germline. With a high prediction accuracy of 0.889 on all positions, it achieves only a 0.031 accuracy on the mutations. It implies that the masking among all positions on the sequence can do accurate predictions of the shared residues but hardly capture the mutation information.

We also conduct AGP and MPP to finetune the baseline model AntiBERT. The pre-training results are shown in Table 8. We can find that without the concatenation of the antibody and its germline, it is difficult to predict the ancestor relationship. It also underperforms than `EATLM` in MPP.

**Negative sampling ratio**    We have tried the ratio of 0.1/0.3/0.5/0.75 and found that this ratio has little influence on performance and convergence speed. As we discussed above, the model can quickly converge for AGP and get an accuracy of 0.99.

**Finetuned Protein Language Models and Larger Architecture**    We pre-train our method with a larger architecture and compare it with ESM-1b, which also has 650M parameters. We also further pre-trained the ESMs to transfer to the antibody field. After that, we evaluate them on the antigen binding and paratope prediction task. The results are shown in Table 9. The result shows that the larger architecture does show an advantage in terms of performance improvement. For antigen binding, ESM-1b has better performance than ESM-1. However, in paratope prediction, it performs worse. In addition, for ESM, fine-tuning the antibody dataset may cause the overfitting problem, leading to a decrease in the performance of all three tasks.

Table 9: The performance of larger models and finetuned protein language models on three tasks. The ESM-1 and ESM-1b models have 85M and 650M respectively. The `EATLM`-large have similar architecture with ESM-1b. 'FT' indicates the model is further finetuned on our antibody pre-training dataset.

| | Antigen Binding | | | Paratope | | |
|---|---|---|---|---|---|---|
| | AUC | F1 | MCC | AUC | F1 | MCC |
| ESM-1 | 0.917+0.001 | 0.854+0.002 | 0.689+0.002 | 0.886+0.009 | 0.669+0.024 | 0.547+0.026 |
| ESM-1 FT | 0.914+0.001 | 0.858+0.001 | 0.695+0.003 | 0.883+0.009 | 0.657+0.031 | 0.534+0.030 |
| ESM-1b | **0.924+0.002** | 0.860+0.002 | **0.707+0.003** | 0.873+0.009 | 0.655+0.042 | 0.524+0.036 |
| ESM-1b FT | 0.916+0.003 | 0.844+0.003 | 0.686+0.004 | 0.869+0.010 | 0.660+0.021 | 0.527+0.015 |
| `EATLM` | 0.922+0.052 | **0.862+0.021** | 0.699+0.027 | 0.887+0.008 | **0.698+0.017** | **0.575+0.024** |
| `EATLM`-large | 0.921+0.004 | 0.854+0.004 | 0.677+0.010 | **0.887+0.007** | 0.685+0.015 | 0.561+0.020 |

## A.6    LIMITATION ABOUT EATLM

First, EATLM doesn't use any 3D structure information during pre-training. As a special subgroup of proteins, antibody structures provide much more information such as geometry than sequences. In the future, recruiting structure information for antibody pre-training may be able to improve the results. However, the data scale available for antibody structure is dramatically less than that of antibody sequences. The largest dataset of antibody structures only contains thousands of 3D high-resolution structures, while the number of antibody sequences is in billions. Using structure prediction methods like AlphaFold may help to bridge the gap between sequences and structures. Second, EATLM requires germline as input during downstream tasks, this will slow down the prediction speed.

## A.7    NEW SARS BINDER DISCOVERY

The main challenge for disease diagnosis is to distinguish the disease-related antibodies from millions of antibody sequences in the individual profile, as stated in Section A.3. Here, with the help of a sequence-level predictor, we can give each sequence a most likely label to help antibody discovery, whose accuracy has been verified by the excellent results on individual prediction, which may accelerate the discovery of new antibody sequences.

**SARS Sequence-level Predictor**    We first train a sequence-level predictor for SARS-CoV-2. The results are shown in Table 10. Compared with Figure 4 in the main text, we find that good results in the sequence-level predictor do not necessarily mean good results in the antibody discovery. It can be mainly affected by the noisy label of the sequence level.

**Figure out SARS Binders**    As shown in Table 3 in the main body, we find 2 true SARS binders and 9 potential binders with the help of `EATLM`. Specifically, we first use our sequence-level predictor to

Table 10: Sequence-level predictor for SARS-CoV-2.

| Sequence-level | SARS | | |
|---|---|---|---|
| | AUC | F1 | MCC |
| No pretrain | 0.894±0.029 | 0.801±0.045 | 0.637±0.078 |
| ESM-1 | 0.903±0.061 | 0.799±0.032 | 0.648±0.093 |
| MSA-1b | 0.914±0.025 | 0.810±0.050 | **0.671±0.080** |
| Ablang-H | 0.915±0.027 | **0.817±0.038** | 0.668±0.068 |
| Ablang-L | 0.893±0.035 | 0.801±0.054 | 0.637±0.099 |
| AntiBERT | **0.916±0.026** | 0.810±0.033 | 0.661±0.060 |
| EATLM | 0.904±0.027 | 0.808±0.035 | 0.643±0.069 |
| EATLM w/o AGP | 0.904±0.029 | 0.808±0.039 | 0.644±0.078 |
| EATLM w/o MPP | 0.901±0.026 | 0.808±0.035 | 0.648±0.069 |
| EATLM w/o MPP & AGP | 0.901±0.026 | 0.807±0.034 | 0.645±0.067 |

get a probability score for each sequence in the SARS dataset. Then we select the sequence with a high-ranked score (the probability > 0.5) and compare them with the public Cov-AbDab database Raybould et al. (2021) [1], which contains data on published/patented antibodies known to bind to SARS-CoV-2 (Raybould et al., 2021). Since the CDR3 fragment in the heavy chain is the most relevant to the binding of antibody and antigen, we calculate the edit distance between the CDR3 fragments in heavy chains (CDR-H3) with those of the known binder and use a threshold of 85% similarity as the sequence identity. 85% Hamming distance for B cell antibody sequence clustering (identify similar B cell antibody sequences responding to the same antigen/epitope) was previously suggested in this paper (Gupta et al., 2017). This method then was widely used for B cell antibody repertoire analysis in different studies (Montague et al., 2021; Wang et al., 2022).

**SARS Binder Analysis**    To provide a more intuitive analysis of the similarity between our predicted antibody and true SARS-CoV-2 binders, we investigate the 3D structure of the true binding antibodies and the mutation site of our predicted sequence on the corresponding structure. High resolution structure of true binding antibody #3 in Table 3 with SARS-CoV-2 are shown in Figure 13 (PDB code: 7N62). The interaction interface between the antibodies and SARS-CoV-2 spike/RBD is shown in Figure 3 in the main body. CDR-H3 were shown in orange. Only one single residue highlighted in red is different between the predicted binder and the true binder. Obviously, these different residues don't localize to the direct binding site and CDR-H3 founding core, suggesting the sequence difference likely will not affect antibody-virus interaction. Furthermore, we found the epitopes of the 11 identified SARS-CoV-2 antibodies cover a wide range of different structures from traditional RBD domain to novel non-RBD epitopes like S2 and NTD as shown in Table 3. This result shows our method enables diverse-epitope antibody discovery.

Table 11: SARS-CoV-2 antibody hit rate with different probability thresholds. 'Total' is the total number of sequences whose probabilities are higher than the threshold. 'Hit' is the number of sequences that meet the similarity requirements with existing binders.

| Threshold | Total | Hit | Hit rate (%) |
|---|---|---|---|
| 0.5 | 13253 | 66 | 0.498 |
| 0.7 | 10227 | 54 | 0.528 |
| 0.8 | 9338 | 49 | 0.525 |
| 0.9 | 8178 | 47 | 0.562 |

**Probability Threshold Sensitivity**    In order to investigate the influence of the threshold used to determine the potential binders, we try different thresholds in Table 11. Here, the probability threshold means that if the sequence predictor gives a probability higher than the threshold for one sequence, it will be viewed as a potential binder. If the predicted binder has a sequence similarity higher than

[1]http://opig.stats.ox.ac.uk/webapps/covabdab/

Figure 13: High-resolution structure of mutation in the predicted binder (AKDQDDAYYYYYYMDV) with the existing binding antibody (AKDQDDGYYYYYYMDV).

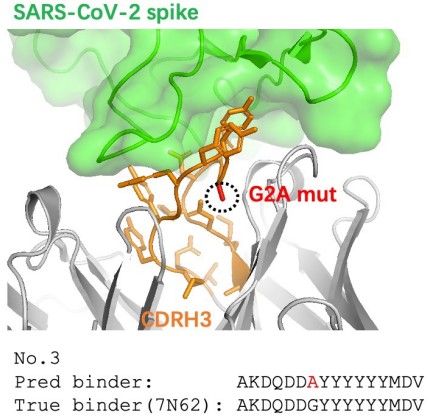

```
No.3
Pred binder:        AKDQDDAYYYYYYMDV
True binder(7N62): AKDQDDGYYYYYYMDV
```

85% with the existing binders in Cov-AbDab, we view it as one hit. As the threshold score increases, the hit rate corresponding increases from 0.528% to 0.562%, indicating that our model may enable priority selection of SARS-CoV-2 antibodies and reduce experimental costs.

**Sequence Similarity Sensitivity**    In previous work, two antibodies with CDR-H3 similarity over 85% can be viewed as similar and have a high probability to share the same functionality. And here we also check the influence on the binder matching of different thresholds of similarity. The results are shown in Figure 14. Here, we fix the probability threshold as 0.5. As we can see, the baselines have similar trends in all thresholds. If we relax the threshold, there will be more matching sequences. However, the predictors will have less advantage over the random order, which indicates that the ranking is less important if we relax the similarity threshold.

**The Potential of New Binder Discovery**    During the training of our sequence-level predictor, we have no reliable ground-truth labels, which means that the model has never known which sequences can bind to SARS in a real-world scenario. However, the model can learn from the noisy data and rank the real SARS binders with high probabilities. Sequence identity of 1 means that the CDR-H3 fragment can be directly found in the Cov-AbDab database, which implies that the sequences have been verified by wet laboratory testing. The other sequences with an identity over 90% are thought to have a similar binding performance to existing binders, indicating that they are promising SARS binders that can help the discovery of therapeutic antibodies for SARS-CoV-2.

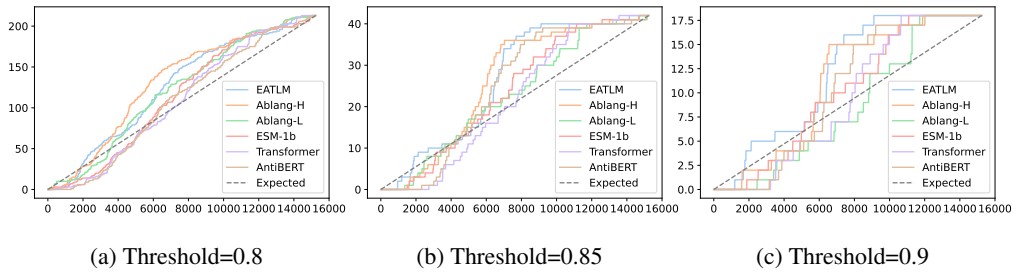

(a) Threshold=0.8          (b) Threshold=0.85          (c) Threshold=0.9

Figure 14: The cumulative count of matching sequences number. The dashed line is the expected results for random order. The x-axis is the sequence number and the y-axis is the cumulative matched sequence number.

A.8 EXTENTED STUDY FOR DISEASE DIAGNOSIS

It would be interesting to see whether our sequence classifier can be used for other applications, such as disease diagnosis. Each human is estimated to maintain about $10^8 - 10^{10}$ distinct antibody sequences, constructing an informative encyclopedia recording the past and present health and disease. Interpreting the pattern of the sequences has already proved useful in disease diagnosis and allows us to assess many infectious diseases without expensive laboratory testing. However, it is difficult to distinguish which antibody sequence from the numerous sequences is responsible for the recognition of the specific antigen, which hinders the discovery of the antibody for diseases (Zaslavsky et al., 2022; Lu et al., 2018; Greiff et al., 2020).

Benefiting from the recent high-throughput sequencing, we can obtain millions of antibody sequences from the individual human. At the same time, we can get a disease label that indicates whether the human is infected by the disease. The main challenge is that we can hardly get the sequence-level disease label that indicates whether the antibody sequence is related to the disease. Thus, we follow the practice of Roskin et al. (2020) to use the individual label as the rough sequence label and train a sequence-level predictor. Then we use this predictor to predict sequences of the individual profile and make the trimmed mean score as the individual score.

We use the same data processing as Antibody Discovery stated in Section A.3. For health/SARS/HIV/Ebola/Allergy/SLE/MS, we set the '*Disease*' field to '*None*', '*Ebola*', '*Allgery*', '*SLE*','*MS*'. Then we obtain 87/133/51/14/12/8/8 patient profiles for each type. We also do 10-cross validation and select sequences with high redundancy.

**Disease Classification** We use all these disease profiles to build the Q7 classification task for disease diagnosis. Previous biological studies mainly use this multi-classification task for disease diagnosis Zaslavsky et al. (2022); Wang et al. (2022), highlighting the discriminatory power among different diseases is important for disease diagnosis. The results are shown in Table 12. We found both PPLM and PALM show comparable results as the randomly initialized model, suggesting the finetuning part plays a more important role and the pre-trained language model cannot help this task.

Table 12: The Q7 disease classification task. 'ACC' is the accuracy rate.

|  | **ACC** |
| --- | --- |
| No pretrain | 0.754±0.023 |
| ESM-1 | 0.747±0.016 |
| ESM-1 FT | **0.762±0.024** |
| MSA-1b | 0.746±0.019 |
| Ablang-H | 0.704±0.033 |
| Ablang-L | 0.702±0.040 |
| AntiBERT | 0.750±0.016 |
| EATLM | **0.756±0.020** |
| EATLM w/o AGP | 0.754±0.020 |
| EATLM w/o MPP | 0.755±0.020 |
| EATLM w/o AGP & MPP | 0.756±0.021 |

**Sequence-level Predictor for Various Disease** As before, we train a sequence-level predictor for each disease. The results are shown in Table 13. Compared with Table 4 in the main text, we find that good results in the sequence-level predictor do not necessarily mean good results in the individual-level predictor. It is mainly due to the trimmed mean we use to get individual-level results, which is a central estimate that is robust to noise labels. Overall, our model has comparable results to other models in terms of sequence prediction with noisy labels and has better results for individual diagnosis.

**Individual-level Predictor for Various Disease** It is observed our evolution-aware EATLM performs the best in the individual-level classifier to determine whether the patient suffering from SARS. Besides, PALMs significantly outperform PPLMs. The results are shown in Table 14.

Table 13: Sequence-level predictor for disease diagnosis.

| Sequence-level | SARS | | | HIV | | |
|---|---|---|---|---|---|---|
| | AUC | F1 | MCC | AUC | F1 | MCC |
| No Pretrain | 0.894±0.029 | 0.801±0.045 | 0.637±0.078 | 0.893±0.081 | 0.622±0.223 | 0.563±0.209 |
| ESM-1 | 0.903±0.061 | 0.799±0.032 | 0.648±0.093 | 0.927±0.039 | 0.739±0.082 | **0.701±0.080** |
| MSA-1b | 0.914±0.025 | 0.810±0.050 | **0.671±0.080** | 0.903±0.064 | 0.680±0.103 | 0.617±0.102 |
| Ablang-H | 0.915±0.027 | **0.817±0.038** | 0.668±0.068 | 0.895±0.058 | 0.684±0.087 | 0.604±0.108 |
| Ablang-L | 0.893±0.035 | 0.801±0.054 | 0.637±0.099 | 0.881±0.072 | 0.668±0.111 | 0.584±0.131 |
| AntiBERT | **0.916±0.026** | 0.810±0.033 | 0.661±0.060 | 0.921±0.046 | 0.742±0.088 | 0.689±0.101 |
| EATLM | 0.904±0.027 | 0.808±0.035 | 0.643±0.069 | 0.930±0.044 | 0.744±0.079 | 0.677±0.105 |
| EATLM w/o AGP | 0.904±0.029 | 0.808±0.039 | 0.644±0.078 | **0.933±0.041** | 0.747±0.077 | 0.679±0.103 |
| EATLM w/o MPP | 0.901±0.026 | 0.808±0.035 | 0.648±0.069 | 0.932±0.037 | **0.749±0.072** | 0.682±0.096 |
| EATLM w/o MPP & AGP | 0.901±0.026 | 0.807±0.034 | 0.645±0.067 | 0.930±0.045 | 0.749±0.074 | 0.678±0.098 |

| | Ebola | | | Allergy | | |
|---|---|---|---|---|---|---|
| | AUC | F1 | MCC | AUC | F1 | MCC |
| No Pretrain | 0.967±0.031 | 0.796±0.115 | 0.787±0.113 | 0.994±0.006 | 0.976±0.017 | 0.958±0.019 |
| ESM-1 | 0.970±0.035 | 0.845±0.098 | 0.836±0.097 | 0.994±0.006 | 0.988±0.006 | 0.975±0.011 |
| MSA-1b | 0.970±0.026 | 0.803±0.103 | 0.799±0.096 | 0.966±0.019 | 0.918±0.037 | 0.835±0.093 |
| Ablang-H | 0.951±0.024 | 0.700±0.087 | 0.691±0.079 | 0.821±0.093 | 0.734±0.091 | 0.459±0.190 |
| Ablang-L | 0.945±0.027 | 0.719±0.110 | 0.710±0.100 | 0.782±0.136 | 0.735±0.104 | 0.381±0.262 |
| AntiBERT | 0.959±0.030 | 0.804±0.110 | 0.795±0.106 | 0.993±0.008 | 0.984±0.009 | 0.970±0.014 |
| EATLM | 0.969±0.036 | **0.848±0.096** | **0.841±0.094** | 0.995±0.006 | **0.988±0.005** | **0.975±0.011** |
| EATLM w/o AGP | 0.967±0.032 | 0.843±0.097 | 0.835±0.093 | 0.996±0.004 | 0.983±0.006 | 0.968±0.007 |
| EATLM w/o MPP | 0.969±0.037 | 0.840±0.094 | 0.828±0.098 | 0.996±0.004 | 0.980±0.005 | 0.961±0.011 |
| EATLM w/o MPP & AGP | **0.970±0.025** | 0.821±0.098 | 0.811±0.091 | **0.996±0.002** | 0.981±0.007 | 0.963±0.009 |

| | SLE | | | MS | | |
|---|---|---|---|---|---|---|
| | AUC | F1 | MCC | AUC | F1 | MCC |
| No Pretrain | 0.994±0.004 | 0.982±0.009 | 0.960±0.012 | 0.841±0.182 | 0.847±0.129 | 0.617±0.378 |
| ESM-1 | 0.992±0.005 | 0.979±0.014 | 0.954±0.021 | 0.879±0.127 | 0.845±0.123 | 0.621±0.358 |
| MSA-1b | **0.998±0.001** | **0.988±0.007** | **0.973±0.011** | 0.853±0.150 | 0.836±0.135 | 0.594±0.392 |
| Ablang-H | 0.994±0.003 | 0.964±0.022 | 0.931±0.021 | 0.836±0.184 | 0.828±0.108 | 0.562±0.353 |
| Ablang-L | 0.995±0.002 | 0.977±0.017 | 0.956±0.021 | 0.846±0.175 | 0.852±0.115 | 0.633±0.351 |
| AntiBERT | 0.994±0.009 | 0.952±0.068 | 0.919±0.096 | **0.893±0.095** | **0.854±0.112** | **0.701±0.214** |
| EATLM | 0.990±0.008 | 0.970±0.018 | 0.939±0.018 | 0.847±0.135 | 0.798±0.156 | 0.572±0.294 |
| EATLM w/o AGP | 0.990±0.007 | 0.962±0.031 | 0.929±0.040 | 0.846±0.132 | 0.799±0.162 | 0.584±0.284 |
| EATLM w/o MPP | 0.981±0.021 | 0.940±0.063 | 0.891±0.086 | 0.809±0.185 | 0.807±0.152 | 0.535±0.381 |
| EATLM w/o MPP & AGP | 0.988±0.010 | 0.952±0.051 | 0.915±0.070 | 0.827±0.159 | 0.798±0.147 | 0.547±0.311 |

Table 14: Individual-level predictor for disease diagnosis.

| Pateint-level | SARS | | | HIV | | |
|---|---|---|---|---|---|---|
| | AUC | F1 | MCC | AUC | F1 | MCC |
| No pretrain | 0.975±0.033 | 0.962±0.024 | 0.902±0.062 | 0.920±0.092 | 0.815±0.117 | 0.776±0.122 |
| ESM-1 | 0.979±0.017 | 0.933±0.032 | 0.824±0.090 | 0.967±0.056 | 0.883±0.107 | 0.849±0.135 |
| MSA-1b | 0.989±0.012 | 0.954±0.033 | 0.887±0.078 | 0.962±0.039 | 0.833±0.068 | 0.789±0.072 |
| Ablang-H | 0.988±0.013 | 0.968±0.024 | 0.920±0.063 | 0.960±0.040 | 0.788±0.134 | 0.744±0.137 |
| Ablang-L | **0.990±0.015** | 0.938±0.039 | 0.843±0.091 | 0.931±0.069 | 0.798±0.131 | 0.742±0.170 |
| AntiBERT | 0.983±0.017 | 0.976±0.022 | 0.938±0.056 | 0.969±0.053 | 0.878±0.112 | 0.850±0.134 |
| EATLM | 0.977±0.028 | **0.983±0.017** | **0.955±0.045** | **0.978±0.054** | 0.914±0.081 | 0.884±0.106 |
| EATLM w/o AGP | 0.983±0.019 | 0.973±0.025 | 0.929±0.066 | **0.978±0.054** | 0.903±0.076 | 0.869±0.099 |
| EATLM w/o MPP | 0.987±0.015 | 0.969±0.024 | 0.921±0.061 | 0.976±0.053 | 0.906±0.092 | 0.880±0.112 |
| EATLM w/o MPP & AGP | 0.986±0.014 | 0.969±0.023 | 0.920±0.061 | 0.973±0.052 | **0.919±0.096** | **0.896±0.117** |

| | Ebola | | | Allergy | | |
|---|---|---|---|---|---|---|
| | AUC | F1 | MCC | AUC | F1 | MCC |
| No Pretrain | 0.994±0.017 | 0.933±0.133 | 0.934±0.132 | 1.000±0.000 | 1.000±0.000 | 1.000±0.000 |
| ESM-1 | 1.000±0.000 | 0.933±0.133 | 0.934±0.132 | 1.000±0.000 | 1.000±0.000 | 1.000±0.000 |
| MSA-1b | 1.000±0.000 | 0.933±0.133 | 0.934±0.132 | 1.000±0.000 | 1.000±0.000 | 1.000±0.000 |
| Ablang-H | 0.944±0.167 | 0.767±0.213 | 0.750±0.254 | 1.000±0.000 | 0.960±0.080 | 0.916±0.169 |
| Ablang-L | 0.961±0.100 | 0.731±0.205 | 0.703±0.277 | 1.000±0.000 | 0.960±0.080 | 0.916±0.169 |
| AntiBERT | 0.978±0.067 | 0.933±0.133 | 0.934±0.132 | 1.000±0.000 | 1.000±0.000 | 1.000±0.000 |
| EATLM | 1.000±0.000 | **0.947±0.111** | **0.944±0.114** | 1.000±0.000 | 1.000±0.000 | 1.000±0.000 |
| EATLM w/o AGP | 1.000±0.000 | 0.933±0.133 | 0.934±0.132 | 1.000±0.000 | 1.000±0.000 | 1.000±0.000 |
| EATLM w/o MPP | 1.000±0.000 | 0.933±0.133 | 0.934±0.132 | 1.000±0.000 | 1.000±0.000 | 1.000±0.000 |
| EATLM w/o MPP & AGP | 0.994±0.017 | 0.933±0.133 | 0.934±0.132 | 1.000±0.000 | 1.000±0.000 | 1.000±0.000 |

| | SLE | | | MS | | |
|---|---|---|---|---|---|---|
| | AUC | F1 | MCC | AUC | F1 | MCC |
| No Pretrain | 1.000±0.000 | 1.000±0.000 | 1.000±0.000 | 0.700±0.200 | 0.900±0.137 | 0.893±0.400 |
| ESM-1 | 1.000±0.000 | 1.000±0.000 | 1.000±0.000 | 0.900±0.200 | 0.933±0.133 | 0.900±0.200 |
| MSA-1b | 1.000±0.000 | 1.000±0.000 | 1.000±0.000 | 0.700±0.400 | 0.893±0.137 | 0.700±0.400 |
| Ablang-H | 1.000±0.000 | 1.000±0.000 | 1.000±0.000 | 0.800±0.245 | 0.893±0.137 | 0.700±0.400 |
| Ablang-L | 1.000±0.000 | 1.000±0.000 | 1.000±0.000 | 1.000±0.000 | 0.960±0.080 | 0.800±0.400 |
| AntiBERT | 1.000±0.000 | 1.000±0.000 | 1.000±0.000 | 1.000±0.000 | 1.000±0.000 | 1.000±0.000 |
| EATLM | 1.000±0.000 | 1.000±0.000 | 1.000±0.000 | 1.000±0.000 | 0.867±0.163 | 0.800±0.245 |
| EATLM w/o AGP | 1.000±0.000 | 1.000±0.000 | 1.000±0.000 | 1.000±0.000 | 0.867±0.163 | 0.800±0.245 |
| EATLM w/o MPP | 1.000±0.000 | 1.000±0.000 | 1.000±0.000 | 1.000±0.000 | 0.827±0.150 | 0.600±0.374 |
| EATLM w/o MPP & AGP | 1.000±0.000 | 1.000±0.000 | 1.000±0.000 | 1.000±0.000 | 0.827±0.150 | 0.800±0.374 |

