# OpenReview forum: "On Pre-training Language Model for Antibody"
_ICLR.cc/2023/Conference — ICLR 2023 poster_

### Official Review · Reviewer_nMMN · 2022-10-22

**Confidence:** 4
**Correctness:** 2
**Technical Novelty And Significance:** 3
**Empirical Novelty And Significance:** 3
**Recommendation:** 5

**Clarity, Quality, Novelty And Reproducibility:**

The paper is not written clearly enough, lacks technical details, and contains many spelling error. It is unclear if the proposed benchmark and methods will be open-sourced.

**Strength And Weaknesses:**

## Strengths
* I am not aware of an existing benchmark specifically for antibodies
* The described loss functions for incorporating the evolutionary relationship of antibodies during pre-training is interesting and new as far as I know

## Weaknesses
* The paper is not written clearly enough. The lack of technical details, unclear definitions such as "Task specificity", and spelling errors make it hard to understand the paper.
* Performance improvements are overall small
* The benchmark contains only five tasks, train/test splits are not justified, and it is unclear if it will be open-sourced. It also does not allow splitting datasets in alternative ways, e.g. by varying the size of the training set or distance to a wildtype sequence.

1) The definition of "task specificity" is unclear and needs to be assessed quantitatively. As a consequence, the conclusion that the proposed loss functions improve performance most on the "most specific" tasks is vague.

2) Please describe the "Evolution-aware antibody pretraining method" more formally by using equations. Phrases such as "The model is made to distinguish the ancestor germline of the antibody by capturing the shared features" are insufficient for understanding the necessary technical details to reimplement the loss function.

3) Please correct spelling and grammatical errors throughout the paper.

4) Please describe how and which hyper-parameters of the proposed model and baseline models were tuned?

5) Please describe how models were fine-tuned and if they were all fine-tuned in the same way.

6) Please compare the number of parameters of baseline models and EATLM (w/o AGP, w/o MPP, AGP & MPP) in table 1. Performance improvements can be due to different numbers of parameters rather than differences in the loss function.

7) Please justify how datasets were split into train/test/eval splits. Sequences of the train and test set can be very similar if, e.g., datasets are split randomly. What does "training/validation/test split of 15,128/3,242/3,242", for example, mean?

8) The benchmark lacks regression tasks to assess the performance of, e.g., continuous binding affinities (10.48550/arXiv.2210.02881).

9) Please cite Li et al (10.48550/arXiv.2210.02881) in the related work section, who recently proposed an antibody benchmark with two tasks.

10) Please describe whether benchmark datasets and baseline models will be open-sourced

11) Table 2: Please separate metrics of different tasks by vertical lines. It is hard to follow which metrics belong to which tasks.

12) Figure 3: The caption is unclear. Does it show a confusion matrix of model predictions vs. ground-truth labels? The performance of which model is shown? How do per-class performances vary across models? Which class is hardest to predict?

13) Figure 4: Also quantify performances by reporting the AUC and alternative ranking metrics such as Spearman's R or NDCG score.

**Summary Of The Paper:**

The paper describes 1) five antibody prediction benchmark tasks, and 2)  two loss functions for pre-training antibody language proteins to incorporate the evolutionary relationship of antibodies during pre-training.

**Summary Of The Review:**

I suggest rejecting the paper since it is not written clearly enough and lacks technical details, which make it hard to understand the proposed methodology and assess benchmark results.

---

> ### Author Response · Authors · 2022-11-14
> **We thank the reviewer for all the valuable questions and suggestions.**
>
> **Q1: The paper is not written clearly enough. The lack of technical details make it hard to understand the paper.**
>
> **A1:** Thank the reviewer for the comment. Please refer to general responses 1. **The main point of our study aims to provide a comprehensive analysis of the performance of protein/antibody pre-trained language models on ATUE benchmark. Due to the space limitation of the main text, we provided pre-training details in Appendix A.5.2 in the original manuscript.** For example, the hyper-parameter necessary to implement the loss function has been shown in Appendix A.5, as mentioned at the beginning of Section 3.3. The framework is further demonstrated in Figure 9. In the revised manuscript, we polished the manuscript and moved some necessary technical details to the main body. The other technical details the reviewer asked including hyper-parameter fine-tuning, number of parameters of baseline models, and data splitting can also be found in Appendix A.5.2.
>
> ---
>
> **Q2: Unclear definitions such as "Task specificity".**
>
> **A2:** Please refer to general response 3.
>
> ---
>
> **Q3: Performance improvements are overall small.**
>
> **A3:** Please refer to general responses 2.
>
> ---
>
> **Q4: The benchmark contains only five tasks, train/test splits are not justified, and it is unclear if it will be open-sourced. It also does not allow splitting datasets in alternative ways, e.g. by varying the size of the training set or distance to a wildtype sequence.**
>
> **A4:** Please refer to general responses 1. We have provided the ATUE dataset in supplementary materials.
>
> ---
>
> **Q5: Please describe the "Evolution-aware antibody pretraining method" more formally by using equations. Phrases such as "The model is made to distinguish the ancestor germline of the antibody by capturing the shared features" are insufficient for understanding the necessary technical details to reimplement the loss function.**
>
> **A5:** We discuss the method in detail in Appendix A.5, as mentioned at the beginning of Section 3.3. We further demonstrate our framework in Figure 9.
>
> ---
>
> **Q6: The benchmark lacks regression tasks to assess the performance of, e.g., continuous binding affinities (10.48550/arXiv.2210.02881).**
>
> **A6:** Thanks for the comments. From a task specificity point of view, we already provide an antibody binding task in ATUE with low antigen specificity. We agree with the reviewer regression task to test antibody binding is valuable for the enrichment of the benchmark. We will add the tasks and experiments to it as soon as these valuable datasets in this paper are released.
>
> ---
>
> **Q7: Please cite Li et al (10.48550/arXiv.2210.02881) in the related work section, who recently proposed an antibody benchmark with two tasks.**
>
> **A7:** Thanks to the reviewer for the suggestion. We have cited this paper in related work. This paper was submitted to arXiv on 5 Oct 2022, a week later than our manuscript submission. These two benchmarks as well as the code haven't been released. We expect to add the tasks in the future if the benchmark is released.
>
> ---
>
> **Q8: Table 2: Please separate metrics of different tasks by vertical lines. It is hard to follow which metrics belong to which tasks.**
>
> **A8:** Thanks to the reviewer for the suggestion. Separate metrics are added.
>
> ---
>
> **Q9: Figure 3: The caption is unclear. Does it show a confusion matrix of model predictions vs. ground-truth labels? The performance of which model is shown? How do per-class performances vary across models? Which class is hardest to predict?**
>
> **A9:** The element $p_{ij}$ of Figure 3 indicates the frequency for the model to predict the antibody in i category to j category. The row is the ground truth and the column is the model prediction. We use the EATLM to predict since it has the best performance in Table 1. The transitional B cell is the most difficult to predict because the prediction accuracy of this class is only 0.406 ($p_{11}$).
>
> ---
>
> **Q10: Figure 4: Also quantify performances by reporting the AUC and alternative ranking metrics such as Spearman's R or NDCG score.**
>
> **A10:** We follow the practice of Zaslavsky et al. (2022) and attach the classification metric AUC, F1 and MCC of the sequence-level classifier in Table 9 in Appendix, and the individual-level classifier in Table 12.

---

> > ### Author Response · Authors · 2022-11-22
> > **Looking forward to hearing your feedback!**
> >
> > Dear reviewer,
> >
> > Thank you so much for taking the time to share your valuable feedback. We have revised the manuscript and performed further analysis to address your concerns. Please see our response for more details. We look forward to hearing more about your thoughts and would be happy to answer more follow-up questions.

---

> > > ### Author Response · Authors · 2022-11-28
> > > **Looking forward to hearing your feedback!**
> > >
> > > Dear reviewer,
> > >
> > > Thank you for taking the time to review our paper. We have polished our paper following your suggestions, especially the technical details and benchmark release. We hope our responses have addressed your concerns raised so far. In case of any unresolved questions or further concerns, please let us know.
> > >
> > > Best wishes,
> > >
> > > Paper 6062 Authors

---

> > > > ### Author Response · Authors · 2022-11-29
> > > > **Any other concerns?**
> > > >
> > > > Dear reviewer,
> > > >
> > > > We have revised the manuscript and performed further analysis to address your concerns. Also, we noticed you have kindly increased the score from 3 to 5. Would you please let us know the remaining concerns you might have? We are always happy to provide further clarification and improve the quality of the work.
> > > >
> > > > Thank you very much for your time!
> > > >
> > > > Best wishes,
> > > >
> > > > Paper 6062 Authors

---

> > > > > ### Comment · Reviewer_nMMN · 2022-11-30
> > > > > **Justification of final score**
> > > > >
> > > > > Dear authors,
> > > > >
> > > > > Thanks for rigorously addressing my comments! I have increased my score from 3 to 5 as you noticed. Not higher since the performance improvement of the proposed EATLM remains small and I find (in agreement with reviewer 5gxi) that what matters is to incorporate evolutionary information in a pre-trained language model "successfully".  I would find it overall more useful for the community if the paper introduced more datasets (benchmark tasks) with alternative ways to split them into training and test sets, e.g. by sequence similarity of distance to a reference protein (similar to the FLIP benchmark). However, I feel that this is out of scope to be addressed in the rebuttal.
> > > > >
> > > > > Best,

---

> > > > > > ### Author Response · Authors · 2022-12-01
> > > > > > **Sincerely thanks to all the valuable questions and suggestions from the reviewer!**
> > > > > >
> > > > > > Dear reviewer,
> > > > > >
> > > > > > We really appreciate your kindness in increasing the score! Thank you again for the valuable response to justify the final score and explanation of your concerns! They are all valuable to increase the quality of the manuscript.
> > > > > >
> > > > > > We want to straightforwardly point out that **our contribution is providing insightful analysis and guidelines for antibody modeling, instead of providing SOTA methods.** In the study, we clearly defined antibody-specific evolution (in a  reverse way from protein) and accordingly provided the first antibody-specificity benchmark. The empirical study showed: (1) The performance of pre-trained language models highly depends on the task specificity; Directly transferring protein language models could be harmful. (2) Incorporation of biological evolution mechanism benefits antibody prediction tasks with high specificity.
> > > > > >
> > > > > > For the concerns about EATLM performance and datasets in the benchmark, please refer to the general response 2 and response to reviewer 5gxi. For data splitting, 10-fold cross-validation is widely used in small-sized biological data. We followed the studies utilizing 10-fold cross-validation [1,2]. Many thanks for proposing distance-based methods used in FLIP. Unfortunately, except for the antibody binding task, which has low specificity, the antibody in the remaining tasks has no reference sequences.
> > > > > >
> > > > > > [1] Edgar Liberis, Petar Velickovi, Pietro Sormanni, Michele Vendruscolo, and Pietro Liò. Parapred: antibody paratope prediction using convolutional and recurrent neural networks. Bioinformatics, 34(17):2944–2950, 2018.
> > > > > >
> > > > > > [2] Derek M Mason, Simon Friedensohn, Cédric R Weber, Christian Jordi, Bastian Wagner, Simon M Meng, Roy A Ehling, Lucia Bonati, Jan Dahinden, Pablo Gainza, et al. Optimization of therapeutic antibodies by predicting antigen specificity from antibody sequence via deep learning. Nature Biomedical Engineering, 5(6):600–612, 2021.
> > > > > >
> > > > > > Sincerely,
> > > > > >
> > > > > > Authors of Paper 6062

---

### Official Review · Reviewer_rBr6 · 2022-10-24

**Confidence:** 4
**Correctness:** 3
**Technical Novelty And Significance:** 2
**Empirical Novelty And Significance:** 4
**Recommendation:** 6

**Clarity, Quality, Novelty And Reproducibility:**

I really appreciated how the paper was written. It provides lots of basic background information on antibodies and discusses these complex topics well. I also really appreciated how much of the exposition was structured in terms of whether tasks are antibody-specific or more general to proteins.

In general, I am a big supporter of papers that contribute new benchmarking setups. These can be used to drive methods research for years. This paper appears to be the first setup for antibody-specific benchmarking.


**Strength And Weaknesses:**

=strengths=
Important application
Well written
Provides a first-of-its kind set of benchmarks for antibody ML
Contributes a new interesting antibody-specific LM model

=Weaknesses=
Some of the tasks in the benchmark are based on small datasets, such that reliably computing differences between ML systems may be difficult.
The covid-19 antibody discovery experiments seem to be a bit forced (see below).


**Summary Of The Paper:**

This paper introduces a first-of-its-kind suite of benchmarking tasks for antibody-specific language models and provides some interesting observations about the behavior of general protein models and antibody-specific models on these tasks. It also introduces a new antibody-specific pretraining objective based on the unique evolutionary process of antibodies.


**Summary Of The Review:**

The paper introduces (1) a new set of benchmarking tasks, (2) benchmarks a number of models from recent papers, and (3) introduces a new antibody-specific model. I feel that (1) and (2) should be adequate for acceptance. A paper that introduces a new benchmark shouldn't be required to introduce a novel model that achieves SOTA on this benchmark. However, it appears that (3) performs slightly better than prior work.

The EATLM model is interesting. It adds two new modeling ideas on top of a baseline non-antibody model. It would have been helpful to provide an ablation that shows how much each of these contributes.

Overall, the performance improvement from the proposed EATLM model is positive, but small. It was hard for me to tell if it was actually significant. How did you obtain the error bars in Table 2? I'm concerned that the test sets are small, yet the error bars are small. I recommend obtaining error bars using bootstrap sampling on the test set.  Similarly, in Fig 4 the y axis is small. How do we know that the differences between the lines aren't just due to chance?

Fig 5 seems like a basic sanity check, not a groundbreaking result. Couldn't you achieve something similar, for example, by doing UMAP on basic sequence alignment distances between pairs of sequences in the dataset?

I didn't fully understand the 'Antibody Discovery' section, as this is far outside of my expertise area. As far as I understand, a classifier was trained on a dataset containing functional  and non-functional antibodies against covid. Then, this model was used to screen a list of candidate antibodies. The top-ranked ones were then labeled as true-positives simply if they have high sequence identity to true known positives. Wouldn't any sort of nearest-neighbor classifier be guaranteed to get extremely high performance on this task, by construction? I don't understand why the results are impressive.


Fine tuning language models for downstream tasks is quite challenging, as there are tons of design choices and hyper-parameters. How do you know that what you did provides a fair comparison across models? Is it a standard approach?

---

> ### Author Response · Authors · 2022-11-14
> **We thank the reviewer for all the valuable questions and suggestions.**
>
> **Q1: Overall, the performance improvement from the proposed EATLM model is positive, but small. It was hard for me to tell if it was actually significant. How did you obtain the error bars in Table 2? I'm concerned that the test sets are small, yet the error bars are small. I recommend obtaining error bars using bootstrap sampling on the test set. Similarly, in Fig 4 the y axis is small. How do we know that the differences between the lines aren't just due to chance?**
>
> **A1:** Thanks to the reviewer for the comments. Please refer to general responses 1. To avoid inductive bias, we conduct **10-fold cross-validation** and report the mean and std. For results in Fig 4, we also trained 10-fold SARS antibody classifiers and used the mean score predicted by the classifiers for evaluation. Consistent to the results shown in Fig4, we can observe similar trends in model performances when different sequence similarity thresholds are applied in the tests, as shown in *Table 13* in *Appendix A.7*.
>
> ---
>
> **Q2: Fig 5 seems like a basic sanity check, not a groundbreaking result. Couldn't you achieve something similar, for example, by doing UMAP on basic sequence alignment distances between pairs of sequences in the dataset?**
>
> **A2:** Indeed, sequence alignment distance between germlines and antibodies can achieve similar results. We thank the reviewer for pointing out that the result shown in Fig 5 is a basic sanity check to test the ability of EATLM to extract the sequence information of antibody. The embedding is correlated with token-level distance of antibody sequences, and nicely can discriminate different germlines. We also added the following sentence at the end of this *section 4.2 Ancestor Germline Visualization: "The visualization provides a sanity check for the ability of EATLM to extract the sequence information of antibody."*
>
> ---
>
> **Q3: I didn't fully understand the 'Antibody Discovery' section, as this is far outside of my expertise area. As far as I understand, a classifier was trained on a dataset containing functional and non-functional antibodies against covid. Then, this model was used to screen a list of candidate antibodies. The top-ranked ones were then labeled as true-positives simply if they have high sequence identity to true known positives. Wouldn't any sort of nearest-neighbor classifier be guaranteed to get extremely high performance on this task, by construction? I don't understand why the results are impressive.**
>
> **A3:** We provided more details for the task in *Appendix A.7*. First of all, we did not use these true-positive binders during the pre-training or fine-tuning. The true binder (CoV-AbDab) sequences were only used for evaluation. What we did is training a sequence-level classifier using patient-level labels. Patient-level labels are so noisy that less than 0.1% of the positive-labeled sequences are true positives. In the dataset, we have 22,000 sequences derived from 133 SARS patients and 87 health. It is known SARS antibodies from different patients show converged sequence patterns allowing the classifier to learn [4].  With the classifier, we can predict a score for each antibody sequence. And the results of *Figure 4* showed that **the antibody sequences with high predictive scores are more consistent with the existing COVID binders than other baselines. The comparison indicates the effectiveness of our classifier.** It implies that the other high-scored antibodies via our classifier, which are different from the existing ones, are also more promising to be *undiscovered and diverse binders*. Therefore, it can accelerate antibody discovery. Especially, when we do not have many true binders for a certain disease but only have noisy antibody profiles, we can use the same way to finetune a classifier and predict scores for them to find new promising binders. Such an approach has been used in biological studies for SARS antibody discovery [5].
>
> ---
>
> **Q4: Fine tuning language models for downstream tasks is quite challenging, as there are tons of design choices and hyper-parameters. How do you know that what you did provides a fair comparison across models? Is it a standard approach?**
>
> **A4:** Many thanks for the questions. Please refer to general response 1.

---

### Official Review · Reviewer_UNrF · 2022-10-24

**Confidence:** 3
**Correctness:** 4
**Technical Novelty And Significance:** 2
**Empirical Novelty And Significance:** 2
**Recommendation:** 6

**Clarity, Quality, Novelty And Reproducibility:**

Clarity: Great

Quality: Good

Novelty: Good

Reproducibility: Easy to reproduce.

**Strength And Weaknesses:**

Strength:
  - This paper is really well-written and easy to follow. The authors provide essential biological and technical backgrounds and clearly state the status, problems, methods, and empirical results.
  - The problem it tries to solve is important, and the authors provide great insights into this problem.
  - The provided benchmark could be helpful for future studies.

Weaknesses:
  - Besides the analysis and insights, the contribution may not be significant enough.
  - From the modeling perspective, this paper just introduced two new training targets besides MLM that leads to slightly better performance compared to baselines such as *Ablang-H*.
  - From the benchmark perspective, providing new datasets or incorporating more existing datasets would make this contribution much more significant.



**Summary Of The Paper:**

This paper provides a comprehensive analysis of Pre-trained Protein Language Models (PPLM) and specific Pre-trained Antibody Language Models on the predictions of different antibody tasks and introduces a new pre-trained method that better utilizes antibody-specific information to achieve a pre-trained antibody language model.

**Summary Of The Review:**

Overall, this paper is of high quality. Considering its technical novelty and empirical performance, I would recommend a weak acceptance.

---

> ### Author Response · Authors · 2022-11-14
> **We thank the reviewer for all the valuable questions and suggestions.**
>
> **Q1: Besides the analysis and insights, the contribution may not be significant enough.**
>
> **A1:** Thank the reviewer for the comment. Please refer to general responses 2.
>
> ---
>
> **Q2: From the modeling perspective, this paper just introduced two new training targets besides MLM that leads to slightly better performance compared to baselines such as Ablang-H.**
>
> **A2:** Please refer to general responses 2.
>
> ---
>
> **Q3: From the benchmark perspective, providing new datasets or incorporating more existing datasets would make this contribution much more significant.**
>
> **A3:** We thank the reviewer for the comment. We agree that the article would benefit from more existing datasets. We attempted to provide more datasets. In *Appendix A.7*, we provided the disease diagnosis task. Antibodies related to three virus infection diseases (SARS/HIV/Ebola), three immune diseases (Allergy/SLE/MS), and health were incorporated into the data. The results were shown in *Table 12*. We found both PPLM and PALM show comparable results as the randomly initialized model, suggesting the data size in the task is so large that model fine-tuning plays a more important role, and the pre-trained language model cannot help this task.
>
> We want to emphasize that antibody datasets are usually scarce clinical data that is rarely publicly released. Our work provides **the first benchmarking tasks for antibody-specific language models**. With the data increasing in this field, we expect we can update our benchmark with more disease diversity and data size in the future.

---

### Official Review · Reviewer_5gxi · 2022-10-30

**Confidence:** 5
**Clarity, Quality, Novelty And Reproducibility:** See above. Novelty is incremental.
**Correctness:** 2
**Technical Novelty And Significance:** 2
**Empirical Novelty And Significance:** 2
**Recommendation:** 6

**Strength And Weaknesses:**

Strength:
1. The authors study the antibody understanding tasks, where antibody is the main and crucial element in drug discovery. The authors propose a new benchmark for the antibody understanding tasks, which contain four specific applications.
2. The authors propose new biological information involved antibody pre-training methods, which improve the understanding of antibody.
3. The authors study different pre-training models for antibody and they have several conclusions.
4. The paper is clear and easy to follow.

Weaknesses:
1. The authors described about the evolution information about the antibody. In their words, the antibody mutation is targeted at the specific objectives, for example to target on the specific antigen. This is somehow questionable, which is a result driven conclusion. Indeed, protein is also randomly mutated, while the last kept ones have specific structures, functions and so on. The differences between antibody mutation and protein mutation is hard to be convinced.
2. The authors propose two new biological information (evolution) based pre-training objectives, which are actually straightforward. Though they are reasonable, as far as I see, the results are hard to say that these two are effective enough. In terms of these pre-training, different reasons may cause the performance change. I would like the authors to provide more details about the pre-training. For example, how to evaluate the pre-training performances. Indeed, the current ways are like multi-task pre-training. This is a little not enough.
3. As for the created benchmark, one question is about the data, the authors mentioned the different specificities of these antibodies. I feel good about this, but the datasets seem not be so good enough. The first two tasks are from the same dataset, also the first affinity prediction is much like the last lack, only specific to covid. Besides, the performances on some tasks are already 0.8-0.9, which seem to be somehow good enough. That's what doubted me about the importance of these tasks.

**Summary Of The Paper:**

This paper studies the different pre-training models for the antibody understanding tasks, propose new methods with biological information for pre-training, and a new antibody understanding benchmark is created. With different study experiments, the authors conclude several observations from different perspectives.

**Summary Of The Review:**

See above.

---

> ### Author Response · Authors · 2022-11-14
> **Thank the reviewer for all the valuable questions and suggestions.**
>
> Please refer to general response for several concerns. Here we answer other questions:
>
> **Q1: The authors described about the evolution information about the antibody. In their words, the antibody mutation is targeted at the specific objectives, for example to target on the specific antigen. This is somehow questionable, which is a result driven conclusion. Indeed, protein is also randomly mutated, while the last kept ones have specific structures, functions and so on. The differences between antibody mutation and protein mutation is hard to be convinced.**
>
> **A1:** Please refer to general responses 3.
>
> ---
>
> **Q2: The authors propose two new biological information (evolution) based pre-training objectives, which are actually straightforward. Though they are reasonable, as far as I see, the results are hard to say that these two are effective enough. In terms of these pre-training, different reasons may cause the performance change. I would like the authors to provide more details about the pre-training. For example, how to evaluate the pre-training performances. Indeed, the current ways are like multi-task pre-training. This is a little not enough.**
>
> **A2:** Please refer to general responses 1 and 2.
>
> ---
>
> **Q3: As for the created benchmark, one question is about the data, the authors mentioned the different specificities of these antibodies. I feel good about this, but the datasets seem not to be so good enough. The first two tasks are from the same dataset, also the first affinity prediction is much like the last lack, only specific to covid.**
>
> **A3:** Thanks for pointing this out. Actually, the first two tasks are from different datasets: Antigen Binding Prediction from Mason et al., 2021 [1], and Paratope Prediction from Liberis et al., 2018 [2]. We have modified the misleading description in the current version. More detailed ATUE information can be found in *Appendix A.3 and A.4*. The first affinity prediction is not for covid, but for HER2 a breast cancer antibody. Here, we want to highlight ATUE benchmark is the first antibody-specific dataset comprising of antibody tasks displaying different scales of antibody specificity, providing a good foundation for model performance evaluation.
>
> ---
>
> **Q4: the performances on some tasks are already 0.8-0.9, which seem to be somehow good enough. That's what doubted me about the importance of these tasks.**
>
> **A4:** The F1 and MCC metric indicates that current performance is not enough for drug discovery. Under the real scenario of drug discovery, the data is imbalanced which brings many challenges for the tasks. For example, there are only a few antibodies that can bind to the antigen (label=1). Therefore, the model can easily obtain high accuracy by predicting all samples as 0. When we look at the F1 metric in Table 1, we can find the results are far away from satisfactory, which means that there is a lot of space to improve. On the application value, the tasks are highly valuable for their close relation to real-world antibody drug developments.

---

> > ### Comment · Reviewer_5gxi · 2022-11-22
> > **Thanks for the rebuttal**
> >
> > Thanks the author for providing the detailed rebuttal.
> > I have replied some notes in the general comments. As for Q4, I would like to further ask, if this is hard and the performance is not enough, why not use other metrics or define other tasks? For example, the regression based method way instead of 0-1 classification? The number (0.8-0.9) gives a somehow good performance in a glance.

---

> > > ### Author Response · Authors · 2022-11-22
> > > **Thanks to the reviewer for the question!**
> > >
> > > For the classification tasks, we used three metrics AUC, F1, and MCC. According to the results evaluated with F1 (the number 0.5-0.8) and MCC (the number 0.5-0.6), we can clearly find the results are far away from satisfactory. For task definition, we were not able to include any regression task because it is very hard to obtain the quantitative label for each antibody sequence in reality, which totally relies on highly expensive wet lab experiments. We hope we can recruit more tasks and update our benchmark in the future when more datasets are released.

---

> > > > ### Comment · Reviewer_5gxi · 2022-11-22
> > > > **Thanks**
> > > >
> > > > Dear Authors,
> > > >
> > > > Thanks for the response. I see interesting views in your discussions (also in general response).
> > > > I hope this paper can attract attention to antibodies, as you hope.
> > > > Good luck with the paper, I will increase the score.

---

> > > > > ### Author Response · Authors · 2022-11-22
> > > > > **Thanks**
> > > > >
> > > > > Dear Reviewer，
> > > > >
> > > > > Thank you so much for all the valuable feedback and suggestions!

---

### Author Response · Authors · 2022-11-14
**General responses (1/3)**

We would like to thank all reviewers for taking the time to carefully read our manuscript and sharing encouraging comments for both our benchmark and analysis.

All reviewers agree that our *first-of-its-kind antibody-specific tasks (ATUE)* are valuable for the community for setting up a benchmark for this field for future studies (by reviewer: 5gxi, UNrF, rBr6, and nMMN). Most reviewers agree that our *paper is well-written* (by reviewers 5gxi, UNrF, rBr6 ) and our pre-training model analysis provides *essential biological and technical backgrounds and great insights into the problem* (by reviewers UNrF, rBr6).

We would like to thank all reviewers for sharing their valuable feedback and suggestions for our original manuscript. We have polished our manuscript and performed new experiments following suggestions from all reviewers.

- On experimental details, all pre-training implementation details have been provided in detail in current Appendix A.5 (Appendix A.4 before).  In particular, we performed 10-fold cross-validations on various pre-training language models to insure the performance improvement is significant and model comparisons are convincing.
- On the contribution/novelty of EATLM, we discuss the point of proposing EATLM in our paper and the contribution of our study.
- On the definition of antibody evolution, we provide a detailed definition of antibody evolution specificity as well as a quantitative analysis of task specificity in Appendix A.1 and A.4.

We have addressed the general comments and summarized the major changes below, and responded to specific points in the individual threads.

**Q1: Experiment details**

The paper lacks experimental details, such as the implementation of pre-training and fine-tuning.

**A1:** We thank the reviewer for the comments. The main point of our study aims to provide a comprehensive analysis of the performance of protein/antibody pre-trained language models on ATUE benchmark. To clarify the comprehensive analysis in the space-limited main text, we only keep the necessary technical information in the main test, while putting the majority of the details of pre-training in *current Appendix A.5.2* (Section A.4 before), as stated in *Section 3.3 Experiment setup* in the original manuscript. In the revised manuscript, we polished the manuscript and moved some necessary technical details to the main body, for example:

(1) On performance evaluation details from reviewer 5gxi, rBr6 and nMMN, we follow the standard training/validation/test split provided by Mason et al., 2021 for Antigen Binding Prediction. We report the average and standard deviations(std) for **three repetitive experiments**. For other tasks that do not have a standard split, we do not provide a fixed split since the dataset size is small. Instead, to approximate the situation where our model would be applied to unseen data, we conduct **10-fold cross-validation** and report the mean and std, as reported in the previous antibody studies (Liberis et al., 2018).

(2) For the pre-training phase from reviewer nMMN, we have provided pre-training details in *Appendix A.5.2*. We conduct the early stop when the loss does not decrease for 20 validation passes (valid every 1000 training iterations). For the fine-tuning phase, all models are fine-tuned with the same hyper-parameters for fair comparison (e.g. lr=3e-5, max epoch = 30). The pre-training performance is shown in *Table 9*.

We have modified the manuscript to specify "Reproduction: We conduct 10-fold validation on paratope prediction, B cell maturation analysis, and antibody discovery. For antigen binding prediction, we conduct three repetitive experiments with different random seeds. We report the average results and the standard derivation. The benchmark and code will be released." These sentences are added in *Section 3.3 Experiment setup*

---

### Author Response · Authors · 2022-11-14
**General responses (2/3)**

**Q2: Contribution of the EATLM**

EATLM only introduces two new training targets and the performance improvement is small.

*"The current ways are like multi-task pre-training. This is a little not enough." (reviewer 5gxi) "From the modeling perspective, this paper just introduced two new training targets besides MLM that leads to slightly better performance compared to baselines such as Ablang-H. Besides the analysis and insights, the contribution may not be significant enough." (Reviewer UNrF)  "This paper propose a new antibody-specific model EATLM. However, it appears that EATLM performs slightly better than prior work." (Reviewer rBr6) "Performance improvements of EATLM are overall small" (Reviewer nMMN)*

**A2:** We thank the reviewers for bringing this to our attention. Here, we want to highlight the aim of the article proposing a new antibody-specific model is not to claim that at this point: the evolutionary aware EATLM approach leads to better representations compared to state-of-the-art antibody pre-training language methods estimated using ATUE. Rather, **we show (i) (for the first time) how antibody-specific evolutionary information can be incorporated in a pre-training language model; and (ii) we used the EATLM approach as an analytic tool to test the idea that an antibody-specific pre-trained language model can improve the performance of pre-training antibody language models (PALMs) on antibody-specific tasks like B cell classification.** We believe this is amply demonstrated by the quantitative results (the small standard deviations(std) for all 10-fold cross-validation experiments) and the ablation studies.

We have modified the manuscript to specify that *"The introduction of two antibody biological mechanisms facilitates PALMs with more antibody-specific features and improves model performance in the task with high antibody specificity. This is the first attempt showing how antibody-specific evolutionary information can be incorporated in a pre-training language model." "The method is used as an analytic tool to investigate the representation ability of antibody evolution-tailored language model."* These sentences are added to the introduction *Our contributions*.

We want to highlight our contribution to the first antibody-specific benchmark ATUE with 4 tasks as well as the comprehensive study which sheds light on the pre-training models for antibody design and discovery.

(1) ATUE benchmark is the first antibody-specific task valuable for the community for setting up a benchmark for future studies. The dataset has been appended as supplementary materials during the paper submission. Both the code and benchmark dataset will be open-sourced later on after the panel decision.

(2) We provide qualitative and quantitative analysis to investigate the performance of current pre-trained language models, and further propose two simple evolution-aware training objectives to verify the effectiveness. As far as we know, it is the first study exploring the pre-training models under antibody-specific views and providing important guidelines to further antibody research.

---

> ### Comment · Reviewer_5gxi · 2022-11-22
> **Partially agree**
>
> Dear Authors,
>
> I think reviewers all agree that the Benchmark is good. The main concern is about the effectiveness of the antibody-specific features currently you used for the antibody pre-trained model. In my view, show how to incorporate this evolutionary information in a pre-trained language model is not a challenging problem to solve, there could be many other ways to incorporate this kind of feature. The important thing is that how we can "successfully" integrate, and I feel the success should also be defined with the performance. Otherwise, it is hard to evaluate the success. Hence, I would suggest this may not be enough.
>
> I thank the authors for the rebuttal, and their attitude and the hard working is what I appreciated a lot.

---

> > ### Author Response · Authors · 2022-11-22
> > **Many thanks for the reviewer 's useful response!**
> >
> > We totally agree with the reviewer that there are many ways to incorporate the evolution information into the pre-training phase. It is definitely valuable to follow the reviewer's suggestion to methodology-wise explore how to incorporate the evolutionary information more effectively, and we leave it as future work. In this work, although the methods we used are simple, the 10-fold cross-validation results shown in Table 2 and Figure 4 show the methods have a significant performance increase over the other pre-trained models on high-specificity tasks. Based on the results, it is reasonable to claim our analytical conclusion that the incorporation of biological evolution is helpful in this paper.

---

### Author Response · Authors · 2022-11-14
**General responses (3/3)**

**"Q3: Definition and quantification of antibody specificity"**

The differences between antibody mutation and protein mutation are hard to be convinced. The definition of "task specificity" is unclear and needs to be assessed quantitatively.

**A3:** Multiple reviewers suggested providing clear definitions and quantitative assessment of "Task specificity" in the analysis. We thank the reviewers for bringing this to our attention. A definition of antibody evolution specificity is originally provided in *Appendix A.1*. As suggested by reviewers, we further recruit protein evolution and compare it with that of antibody's. As shown in *Figure 7A*, protein evolve to have random mutations, and the **conserved amino acids (none mutated) determine protein's structure and function**. On the other hand, antibodies evolve from hundreds of thousands ancestor sequences so-called germline. To bind dozens of millions of diverse antigens, antibodies need to mutate from the ancestor germline sequences to gain new functions. Therefore, the **non-conserved amino acids (mutated ones) play important roles in antibody's structure and function**.

As suggested by the reviewers, it is important to include statistical significance tests relative to the antibody-specific features in antibody functional tasks we proposed in the ATUE benchmark. Here, *we define two antibody evolution specificity features: (i) ancestor germlines; (ii) the mutated amino acids from germlines.* To quantitatively assess the "Task specificity", we implemented statistical significance tests of these two features against different labels of downstream tasks in ATUE. The analysis is now summarized in *Table 7* In *Appendix A.4*. Generally, it is clearly shown that ATUE benchmark comprises antibody tasks displaying different scales of antibody specificity, providing a good opportunity for later model evaluation.

Table 7. Task specificity. Summary of the statistical significance test of two antibody-specific features for different tasks in the ATUE benchmark.

|Tasks|Germline Usage (p-value)|Mutation Numbers (p-value)|Specificity|
| - | - | - | - |
|Antigen Binding|Nan|Nan|low|
|Paratope Prediction|0.296|0|medium|
|B cell classification|0|0|high|
|SARS antibody discovery|0|0|high|

---

> ### Comment · Reviewer_5gxi · 2022-11-22
> **Interesting discussion**
>
> Dear Authors,
>
> I find the discussions (definitions) you provided is interesting, which somehow clarifies my doubt. However, I would like to have more discussions.
> We all agree that the sequences determine the structure, and then the function. In what you provided, co-evolution parts of protein keep the protein structure, does it mean the mutations of other places are not importantly (also no impact to the function), hence the structure would not change a lot if other places are mutated?
> It seems that protein and antibody are totally reverse in these fields.

---

> > ### Author Response · Authors · 2022-11-22
> > **Many thanks for the reviewer 's inspiring discussions!**
> >
> > During evolution, many different mutations can be observed on the non-conserved positions from different homolog proteins, which all share similar structures and functions. Therefore, **yes, the non-conserved (mutated) positions on proteins are not important for their structure maintenance. And mutations on these sites would not change the structure and function a lot [1].**
> >
> > We totally agree with the reviewers' point that proteins and antibodies are reverse in this aspect, which subsequently brings huge differences in the strategy we should use to model antibodies from that of proteins. Unfortunately, this field obviously hasn't been aware of this reverse principle. Until recently, many studies explored antibodies by transferring protein language models and neglecting antibody specificity [2,3]. **We hope our analysis could provide important and prompt guidance for this field in model building and task selection in the future.**
> >
> > Reference:
> >
> > [1] David de Juan, Florencio Pazos and Alfonso Valencia. Emerging methods in protein co-evolution. Nature Review Genetics, 2013.
> >
> > [2] Inyoung Kim, Sang Yoon Byun, Sangyeup Kim, Sangyoon Choi, Jinsung Noh, Junho Chung, and Byung Gee Kim. Analysis of b-cell receptor repertoires in covid-19 patients using deep embedded representations of protein sequences. bioRxiv, 2021.
> >
> >
> > [3] Maxim E Zaslavsky, Nikhil Ram-Mohan, Joel M Guthridge, Joan T Merrill, Jason D Goldman, Ji-Yeun Lee, Krishna M Roskin, Charlotte Cunningham-Rundles, M Anthony Moody, Barton F Haynes, et al. Disease diagnostics using machine learning of immune receptors. bioRxiv, 2022.

---

### Author Response · Authors · 2022-11-14
**Reference**

[1] Derek M Mason, Simon Friedensohn, Cédric R Weber, Christian Jordi, Bastian Wagner, Simon M Meng, Roy A Ehling, Lucia Bonati, Jan Dahinden, Pablo Gainza, et al. Optimization of therapeutic antibodies by predicting antigen specificity from antibody sequence via deep learning. Nature Biomedical Engineering, 5(6):600–612, 2021.

[2] Edgar Liberis, Petar Veliˇckovi´c, Pietro Sormanni, Michele Vendruscolo, and Pietro Liò. Parapred: antibody paratope prediction using convolutional and recurrent neural networks. Bioinformatics, 34(17):2944–2950, 2018.

[3] Marie Ghraichy, Valentin von Niederhäusern, Aleksandr Kovaltsuk, Jacob D Galson, Charlotte M Deane, and Johannes Trück. Different b cell subpopulations show distinct patterns in their igh repertoire metrics. ELife, 10:e73111, 2021.

[4] Jacob D Galson, Sebastian Schaetzle, Rachael JM Bashford-Rogers, Matthew IJ Raybould, Aleksandr Kovaltsuk, Gavin J Kilpatrick, Ralph Minter, Donna K Finch, Jorge Dias, Louisa K James, et al. Deep sequencing of b cell receptor repertoires from covid-19 patients reveals strong convergent immune signatures. Frontiers in immunology, 11:605170, 2020.

[5] Yiquan Wang, Meng Yuan, Huibin Lv, Jian Peng, Ian A Wilson, and Nicholas C Wu. A large-scale
systematic survey reveals recurring molecular features of public antibody responses to sars-cov-2.
Immunity, 2022.

---

### Author Response · Authors · 2022-11-16
**Any questions before the end of the discussion period?**

We would like to thank again all reviewers.

Please let us know if there are additional questions or concerns before the end of the discussion period. We would be happy to discuss or address any additional comments.

---

### Author Response · Authors · 2022-11-28
**Is further clarification needed?**

Dear reviewers and area chairs,

Thank you for taking the time to review our paper. We hope our responses have addressed your concerns raised so far. In case of any unresolved questions or further concerns, please let us know. We are always happy to provide further clarification and improve the quality of the work.

Thank you so much for your time!

---

### Decision · Program_Chairs · 2023-01-20

**Decision:**

Accept: poster

**Justification For Why Not Higher Score:**

Due to the weaknesses pointed out by the reviewers, I think the paper could have been stronger.  Hence, poster is the right decision.

**Justification For Why Not Lower Score:**

Please see above.  I believe that the paper should be accepted.

**Metareview: Summary, Strengths And Weaknesses:**

I found this paper to be quite interesting and thought that it can be a nice contribution to the ICLR community (although this is my opinion as a researcher and AC not too attached to the community focused on protein modeling using LM techniques).  The paper seeks to answer the following research questions, per the authors' text: (1) How do pre-trained language models perform in antibody tasks with different specificity? (2) How many benefits will the model gain if we introduce the specific biological mechanism to the pre-training process? (3) Do the learned antibody pre-trained representations make sense in real-world antibody problems, like drug discovery and immune process
understanding?  In addition, the authors present a new benchmark to facilitate a study on these questions titled AnTibody Understanding Evaluation (ATUE).

Strengths:  The authors study an antibody understanding task using pretrained LMs, they check if specific biological information, when added to the pretraining methods can improve understanding and they also work on a benchmark as mentioned above.  The reviewers agree that the authors put in a lot of work in making this line of work accessible to the readers.  The authors present extensive experiments on several tasks.

Weaknesses:  There is some criticism from the reviewers that the presented methods may not be very novel.  Several other criticisms regarding data quality etc. were brought up that the authors discussed during the review period.  Adding new datasets to the benchmark could also have been more impactful.

While I agree with the weaknesses, I do think that this work can result in a nice connection with the NLP community who are building LMs for general language tasks and the benchmark could be suitable for a new fork in LM research.  Hence, I think it will add value to the ICLR conference.

**Note From Pc:**

if the above contains the word "oral" or "spotlight" please see: "oral" presentation means -> notable-top-5% and "spotlight" means -> notable-top-25%. As stated in our emails, we are disassociating presentation type from AC recommendations